# Sex-specific splicing occurs genome-wide during early *Drosophila* embryogenesis

Mukulika Ray[1†], Ashley Mae Conard[2†], Jennifer Urban[3],
Pranav Mahableshwarkar[1,2], Joseph Aguilera[1], Annie Huang[1],
Smriti Vaidyanathan[1,2], Erica Larschan[1*]

[1]MCB department, Brown University, Providence, United States; [2]CCMB department, Brown University, Providence, United States; [3]Biology department, Johns Hopkins University, Baltimore, United States

**Abstract** Sex-specific splicing is an essential process that regulates sex determination and drives sexual dimorphism. Yet, how early in development widespread sex-specific transcript diversity occurs was unknown because it had yet to be studied at the genome-wide level. We use the powerful *Drosophila* model to show that widespread sex-specific transcript diversity occurs early in development, concurrent with zygotic genome activation. We also present a new pipeline called time2Splice to quantify changes in alternative splicing over time. Furthermore, we determine that one of the consequences of losing an essential maternally deposited pioneer factor called CLAMP (chromatin-linked adapter for MSL proteins) is altered sex-specific splicing of genes involved in diverse biological processes that drive development. Overall, we show that sex-specific differences in transcript diversity exist even at the earliest stages of development..

## Editor's evaluation

In this manuscript, the authors describe the earliest differences in sex-specific splicing in *Drosophila* embryos or any animal for that matter. Based on solid data, they report the important finding that differences arise already during the first few hours of embryogenesis and that a maternally-deposited pioneer transcription factor contributes to generating these differences. The authors also provide a bioinformatics pipeline to analyze splicing over time.

**\*For correspondence:**
erica_larschan@brown.edu

[†]These authors contributed equally to this work

**Competing interest:** The authors declare that no competing interests exist.

## Introduction

One of the greatest challenges in modern biology is understanding the mechanism and significance of widespread transcript diversity between sexes and different developmental stages, tissues, and cell types. Alternative splicing (AS), a mechanism of selective inclusion or exclusion of introns and exons, drives widespread transcript diversity (*Aanes et al., 2013*; *Revil et al., 2010*). In addition to basic development and physiology, transcriptome diversity is critical for disease biology, especially in neurodegenerative diseases and developmental disorders that often show sex or tissue-specific differences in progression and severity (*Mayne et al., 2016*; *Ober et al., 2008*; *Wang and Cooper, 2007*; *Faustino and Cooper, 2003*). Across species, precise regulation of genes to produce specific splice variants is critical for all developmental decisions, including sex determination. A key to understanding how transcript diversity drives biological processes lies in the events that shape the initial few hours of an organism's existence.

During early development, protein and RNA deposited by the mother into the embryo shape early embryonic milestones across metazoans (*Schulz et al., 2015*; *Schulz and Harrison, 2019*). Initially, cell number increases, followed by cellular differentiation into specific cell types. Sexual identity is

then established, driving the fundamental physiological differences between sexes. However, whether maternally deposited proteins and RNAs influence transcript variation remains poorly understood. Moreover, maternal factors are often essential regulators that can have a lasting impact on gene regulation later in the life of an organism. Thus, it is essential to define the influence of maternal factors on transcriptome diversity during the early stages of embryonic development. Therefore, the key question is: How does the presence of maternally deposited products influence transcript diversity including sex-specific splice variants?

The *Drosophila* embryo is an excellent tool in studying the role of maternally deposited proteins and RNA in early development as it is easy to perform genetic manipulations to remove maternal factors to define how they regulate splicing and transcription. Also, embryos can be sexed before zygotic genome activation due to our recent application of a meiotic drive system (*Rieder et al., 2017*). During *Drosophila* embryogenesis, zygotic genome activation (ZGA) occurs shortly after the first 2 hr of development. Concurrently, maternal transcripts gradually decrease in abundance, and zygotic transcription increases, a process called the MZT (maternal to zygotic transition). ZGA starts approximately 80 min after egg laying and most maternal transcripts are degraded by 180 min after egg laying (*Artieri and Fraser, 2014*). Even at these early stages of development, AS generates multiple transcript isoforms resulting in transcript diversity. Although the earliest genes transcribed from the zygotic genome are mainly intron-less, approximately 30% of early zygotic transcripts do have introns (*De Renzis et al., 2007*; *Guilgur et al., 2014*). Furthermore, genes involved in sex determination have introns and use AS to drive male versus female-specific development (*Förch and Valcárcel, 2003*). Hence, during early embryonic development, AS is important for shaping cell and tissue-specific transcriptomes and essential for sexual differentiation. However, it was not known: (1) how early in development widespread sex-specific transcript diversity occurs at a genome-wide level and (2) whether the loss of maternally deposited factors alters sex-specific AS early during embryogenesis.

Several lines of evidence led us to hypothesize that the maternally deposited TF CLAMP (chromatin-linked adapter for MSL proteins) is a good candidate to study how the loss of maternally deposited TFs modulates sex-specific AS: (1) mass spectrometry identified association with 33 RBPs on chromatin, 6 of which regulate AS *Urban et al., 2017c*; (2) CLAMP is bound to approximately equal numbers of intronic regions as promoter regions *Kaye et al., 2018*; (3) many CLAMP binding sites evolved from intronic polypyrimidine tracts *Quinn et al., 2016*; (4) maternal CLAMP is essential for viability in both males and females suggesting a role beyond male-specific dosage compensation (*Duan, 2020*).

First, we defined all of the sex-specifically spliced (SSS) isoforms early during development genome-wide which has never been determined in any species. Although sex-specific isoforms have been identified for several key genes in early development (*Lott et al., 2014*; *Telonis-Scott et al., 2009*; *Paris et al., 2015*; *Lott et al., 2011*), a comprehensive analysis of all SSS isoforms in early embryos was not available. Across species, genome-wide SSS data is only available in adult tissues including ovary, testis, and brain (*Telonis-Scott et al., 2009*; *Gibilisco et al., 2016*). Therefore, we used a meiotic drive system to sex embryos at two time points surrounding MZT (first 4 hr of embryonic development: 0–2 hr and 2–4 hr), and computationally defined all SSS isoforms. We measured AS using time2splice, a new pipeline we developed based on the SUPPA2 algorithm (*Trincado et al., 2018*), that provides additional modules to integrate time and sex as variables (https://github.com/ashleymaeconard/time-2splice; copy archived at *Ray, 2023*).

Next, we analyzed the effects of depleting the maternal transcription factor CLAMP on sex-specific AS during the first few hours of development to understand whether a maternal pioneer factor is required for early embryonic transcriptome diversity. We identified male-specific and female-specific genes involved in development whose splicing required maternal CLAMP. Also, we observed that CLAMP-dependent SSS genes regulate different biological processes in females and males, even as early as when the embryo is activating the zygotic genome. Overall, we demonstrate that sex-specific transcriptome variation is established very early in development and the loss of the maternally deposited TF CLAMP alters SSS of a group of genes that regulate both AS and development. Furthermore, we introduce time2splice, a pipeline to identify alternatively spliced isoforms and how they change over time and in different sexes which can also integrate chromatin localization data of potential splicing regulators.

## Results

### Sex-specific AS is present at the earliest stages of *Drosophila* development

To define when SSS begins during development, we analyzed RNA-sequencing data that we generated from male and female embryos at two-time points: 0–2 hr (pre-MZT) and 2–4 hr (post-MZT) (*Rieder et al., 2017*) (#GSE102922). We were able to produce homogeneous populations of male or female embryos prior to ZGA using a novel meiotic drive system that produces sperm with either only X or only Y chromosomes in which SSS had never been analyzed (*Rieder et al., 2017*). Next, we quantified AS in these samples using a new pipeline that we developed for this analysis and made publicly available called time2splice (https://github.com/ashleymaeconard/time2splice; copy archived at *Ray, 2023*). Time2splice implements the commonly used SUPPA2 algorithm (*Trincado et al., 2018*) to identify splice variants and provides additional modules to integrate time, sex, and chromatin localization data (Materials and methods) (*Figure 1—figure supplement 1*). SUPPA2 measures the PSI (percent spliced in) for each exon and calculates the differential AS between samples, reported as ΔPSI (*Trincado et al., 2018*). Therefore, SUPPA2 is specifically designed to identify AS events.

From our RNA-seq data, we used time2splice to analyze 66,927 exons associated with 17,558 genes and classified the AS events into one of seven classes (diagrammed in *Figure 1A*). We found that 16–18% of the exons are alternatively spliced in early embryos (*Figure 1B*) and fall into one of the seven AS classes (*Figure 1C–D*). Of these seven classes, AF (alternative first exon) is the most common type, constituting almost one-fourth of total AS (~24–26%), and AL (alternative last exon) is the least common type (~3%). The AS transcript distribution across classes was similar between the two time points and the two sexes (*Figure 1B–D*). Next, we asked which type of AS is most affected by depleting maternal CLAMP using our validated RNAi method. The overall distribution of transcripts into the seven AS classes remains mostly unaffected in the absence of maternal CLAMP. However, at the 0–2 hr (pre-MZT) stage, loss of maternal CLAMP results in a more substantial decrease in mutually exclusive exon (**MXE**) splicing in both males and females compared with all of the other types of splicing (**males**: p-value <3.21e-21; **females**: p-value <6.26e-87 Chi-squared test) (*Figure 1D*). At the 2–4 hr/post-MZT stage, only male embryos have a significant percentage of MXE splicing events mis-regulated in the absence of maternal CLAMP (p-value <1.95e-137 Chi-squared test) (*Figure 1D*). Therefore, the depletion of maternal CLAMP mis-regulates AS and has a stronger effect on MXE splicing than other types of splicing.

During MXE splicing, one isoform of the transcript retains one of the alternative exons and excludes another exon, which is retained by another isoform (schematic in *Figure 1A*). Interestingly, MXE AS occurs in many transcripts that encode components of the sex determination pathway (*Brooks et al., 2015*). Furthermore, CLAMP has a sex-specific role in dosage compensation (*Urban et al., 2017a*; *Rieder et al., 2019*). Therefore, we defined SSS events in the early embryo. We identified SSS events in 0–2 hr embryos (pre-MZT) (*Figure 2—figure supplement 1A*, N=92) and in 2–4 hr embryos (post-MZT) (*Figure 2—figure supplement 1B*, N=138) and categorized them as **known SSS** events. Overall, we determined that sex-specific AS occurs earlier in development than ever shown previously in any species.

### Depleting maternal CLAMP alters sex-specific AS in early *Drosophila* embryos

We hypothesized that the loss of CLAMP alters sex-specific AS in early embryos for the following reasons: (1) CLAMP is a maternally deposited pioneer transcription factor with sex-specific functions that is enriched at intronic regions in addition to promoters *Kaye et al., 2018*; *Duan et al., 2021*; (2) proteomic data identified a physical association between spliceosome components and CLAMP *Urban et al., 2017c*; and (3) CLAMP binding sites evolved from polypyrimidine tracts that regulate splicing (*Quinn et al., 2016*). We tested our hypothesis in early stage sexed embryos by measuring differences in AS from RNA-seq data generated from male and female 0–2 hr/pre-MZT and 2–4 hr/post-MZT embryos with and without maternal CLAMP (*Rieder et al., 2017*). The maternal triple driver GAL4 (*MTD-GAL4*) was used to drive *UAS-CLAMPRNAi[val22]* which strongly reduces maternal CLAMP levels as validated by qPCR and western blot conducted in parallel with mRNA-seq data collection (*Rieder et al., 2017*).

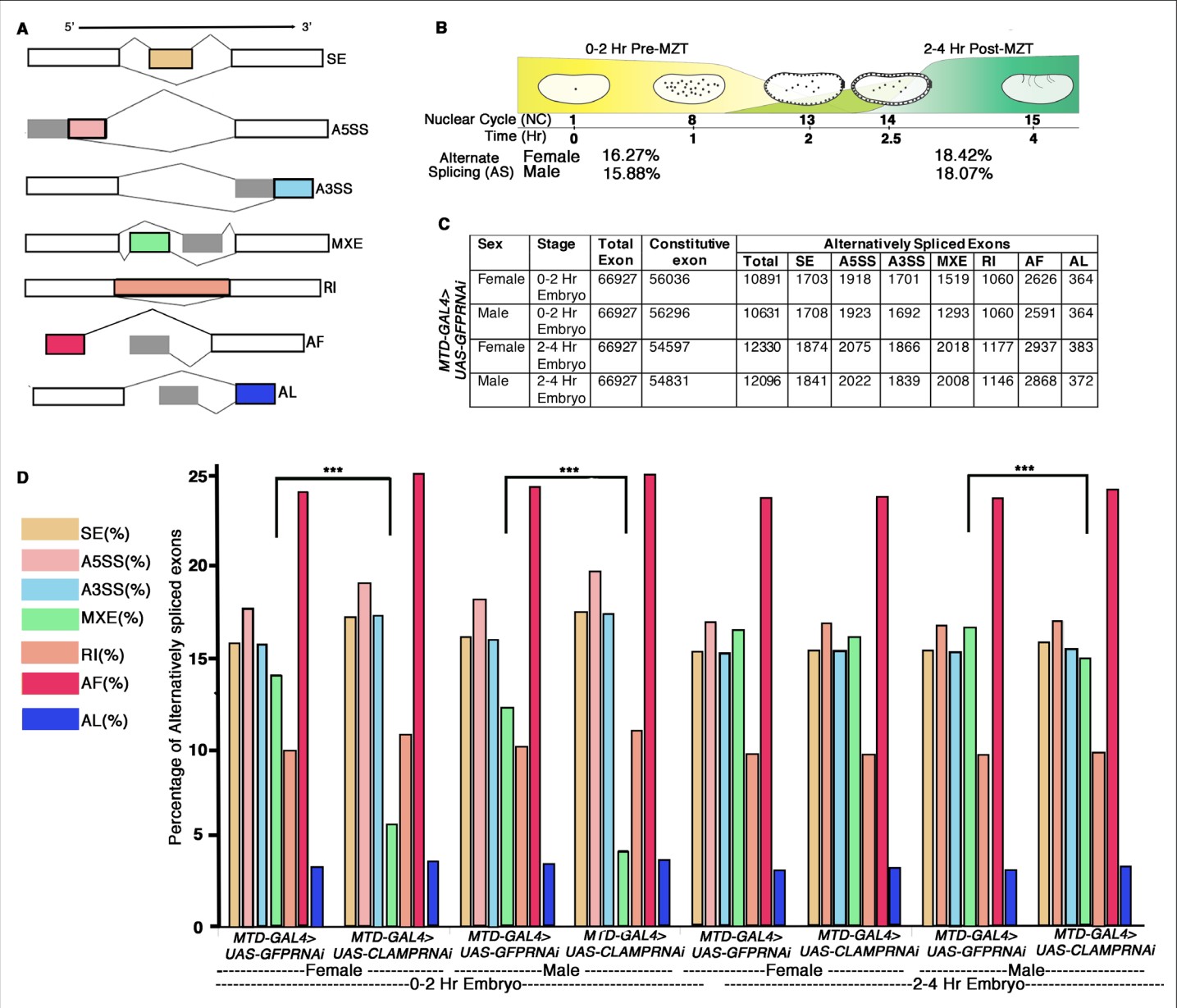

**Figure 1.** Alternative splicing (AS) during early *Drosophila melanogaster* embryonic development. (**A**) Schematic diagrams showing seven different types of AS. The constitutive exons are depicted as white rectangles, whereas the alternatively spliced exons are in shades of gray and different colors rectangles according to type of AS. (**B**) Percentage of genes with AS in male and female early *Drosophila* embryos at the 0–2 hr/pre-MZT (maternal to zygotic transition) and 2–4 hr/post-MZT stages. (**C**) Table showing the number of exons in each AS category in control sexed embryos at the 0–2 hr/pre-MZT and 2–4 hr/post-MZT stages. (**D**) Bar plot showing the distribution of different types of AS (colored according to **A**) at 0–2 hr/pre-MZT and 2–4 hr/post-MZT for female and male embryos in the presence (*MTDGAL4>UAS-GFPRNAi*) and absence (*MTDGAL4>UAS-CLAMPRNAi*) of maternal CLAMP (chromatin-linked adapter for MSL proteins). A Chi-square test was performed to determine if there is a significant difference between the percentage of each type of AS including mutually exclusive exon (MXE) splicing (black bar) in the presence vs. absence of CLAMP in each class of sample: female and male 0–2 hr/pre-MZT and 2–4 hr/post-MZT embryos. Statistically significant differences (p<0.001 marked by ***) were found between categories connected by solid black lines.

The online version of this article includes the following source data and figure supplement(s) for figure 1:

**Source data 1.** Source data for *Figure 1C*.

**Figure supplement 1.** Schematic diagram describing each step-in sequential order performed by the time2Splice pipeline.

First, we asked whether CLAMP alters AS and we found 200–400 transcripts at which AS requires CLAMP, based on the time point and sex (*Figure 2—figure supplement 1C–F* and *Figure 2A and B*). To determine whether CLAMP-dependent AS events are enriched for SSS events, we first identified all of the CLAMP-dependent AS events in female (*Figure 2—figure supplement 1C, D*) and in male (*Figure 2—figure supplement 1E, F*) 0–2 hr and 2–4 hr embryos (Materials and methods). We measured AS using an exon-centric approach to quantify individual splice junctions by measuring PSI for a particular exon using the established SUPPA algorithm within the time2Splice pipeline (*Trincado et al., 2018*). Exon inclusion is represented as positive PSI, and exon exclusion events are defined as negative PSI (equation in Materials and methods). By comparing the CLAMP-dependent AS events in females and males, we identified CLAMP-dependent SSS events in female and male 0–2 hr and 2–4 hr embryos (*Figure 2A and B* and *Supplementary file 1a-h*).

When we measured the percentage of total alternatively spliced and SSS transcripts that require CLAMP in males and females at both pre- and post-MZT stages, we found that while only 2–3% of total AS exons are CLAMP-dependent, ~30–60% of SSS exons are CLAMP-dependent (*Figure 2A*). Therefore, the function of CLAMP in AS is highly enriched for SSS events. We then divided all CLAMP-dependent AS events into two categories: (1) SSS events and (2) non-SSS events (*Figure 2B*).

Next, we subdivided the CLAMP-dependent SSS events into the following subclasses: (1) **known** SSS events: female-specific and male-specific splicing events at 0–2 hr and 2–4 hr embryo stages that are CLAMP-dependent (p<0.05) (*Figure 2A–B*); (2) **new** SSS events: splicing events that occur only in the absence of CLAMP and not in control samples (*Figure 2B*). By calculating ΔPSI in these subclasses, we identified widespread CLAMP-dependent SSS, especially in female embryos (*Figure 2B*). Interestingly, the majority of CLAMP-dependent SSS events are **new** aberrant SSS events that did not occur in the presence of maternal CLAMP (~70%) (*Figure 2C*).

To define the magnitude of the effect of CLAMP on splicing, we compared the ΔPSI for known and new SSS events between female and male samples (*Figure 2—figure supplement 2*). We found that although more splicing events/transcripts show CLAMP-dependent splicing in females (~150–250) than males (~100) (*Figure 2B* and *Supplementary file 1a-h*), post-MZT, CLAMP-dependent exon inclusion was significantly enriched in male **new** SSS transcripts compared to their female-specific counterparts (*Figure 2—figure supplement 2*). Thus, in the absence of CLAMP, new aberrant SSS isoforms are generated.

During the first few hours of their development, *Drosophila* embryos have predominantly maternal transcripts. Therefore, we asked whether CLAMP-dependent female and male specifically spliced genes are maternally deposited or zygotically transcribed. We compared our list of CLAMP-dependent SSS genes with known maternally expressed genes that are consistent across multiple previous studies (*Atallah and Lott, 2018*; *Kwasnieski et al., 2019*). We found very low levels of overlap with maternally deposited transcripts (*Figure 2D*) even in the 0–2 hr embryo stage, consistent with ZGA starting at approximately 80 min after egg laying (*Artieri and Fraser, 2014*). Therefore, most of the SSS genes we observed are likely to be zygotic transcripts, consistent with the function of CLAMP as a maternally deposited factor acting in the early embryo (*Duan, 2020*).

To understand the classes of genes whose splicing requires CLAMP, we performed Gene Ontology (GO) analysis. Our analysis showed that pre-MZT (0–2 hr), female SSS genes (N=119) are primarily TFs and factors that regulate splicing (*Figure 2E*). Therefore, in females the loss of CLAMP alters the splicing of genes that can regulate the transcription and splicing of other genes. In contrast, the male specifically spliced pre-MZT genes (N=96) are not enriched for any specific biological function or process, likely due to the small number of genes in the gene list. At the post-MZT stage in both sexes (female: N=207; male: N=104), the loss of CLAMP mis-regulates splicing of genes that drive development including organogenesis, morphogenesis, cell proliferation, signaling, and neurogenesis (*Figure 2E*).

In order to validate our genomic splicing analysis from time2splice for individual target genes (*Figure 2E*), we randomly selected eight genes at which we determined that CLAMP regulates splicing and measured AS using qRT-PCR or RT-PCR (*Figure 2—figure supplement 3*). Our RT-PCR results indicate that we are able to validate the function of CLAMP in regulating splicing of genes which we identified genomically with time2splice (*Figure 2—figure supplement 3*). We summarized the functions of the validated target genes at which splicing is regulated by CLAMP (*Supplementary file 2*). *iab4*, one of the target genes that we validated, has known functional links to CLAMP because CLAMP

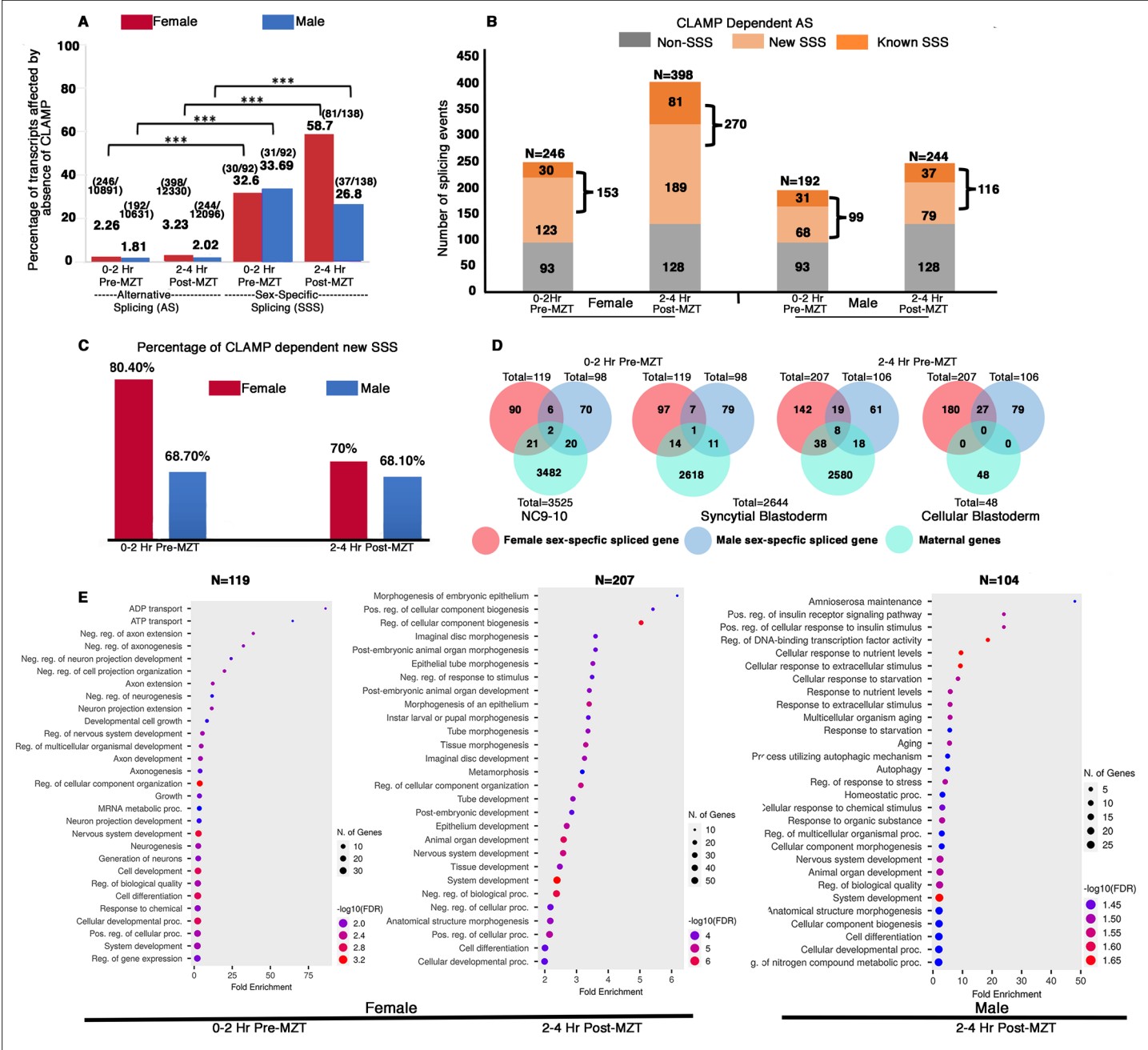

**Figure 2.** Maternal CLAMP (chromatin-linked adapter for MSL proteins) regulates sex-specific alternative splicing (AS) during early embryonic development. (**A**) Bar graph showing the percentage of transcripts (raw values noted at the top of each bar) compared with total AS events or sex-specific splicing (SSS) events within parentheses listed at the top of each bar: number of splicing events regulated by CLAMP/total number of splicing events. We quantified transcripts whose splicing is regulated by maternal CLAMP at the 0–2 hr/pre-MZT (maternal to zygotic transition) and 2–4 hr/post-MZT stages in females (red bars) and males (blue bars). A Fisher's exact test was performed with significance shown at p<0.001. (**B**) Bar plot showing the total number of splicing events undergoing CLAMP-dependent AS (N) in females and males at 0–2 hr/pre-MZT and 2–4 hr/post-MZT embryonic stages. Alternatively, spliced genes are divided into non-sex-specific (gray) and sex-specific (orange shades) sub-categories. CLAMP-dependent female and male SSS genes are divided into known (sex-specific in control samples: darker orange) and new (sex-specific only after depleting CLAMP: lighter orange) sub-categories identified from 0 to 2 hr/pre-MZT and 2–4 hr post-MZT/embryos. (**C**). Percentage of new female (red) and male (blue) CLAMP-dependent SSS genes in 0–2 hr/pre-MZT and 2–4 hr/post-MZT embryos that were not identified as different between males and females in control samples. (**D**). Female (red) and male (blue) CLAMP-dependent SSS genes compared with maternal genes (green, NC9-10 stage, N=3525; syncytial blastoderm stage, N=2644; cellular blastoderm stage, N=48) at 0–2 hr/pre-MZT (female, N=119 and male, N=98) and 2–4 hr/ post-MZT stages (female, N=207 and male, N=106). (**E**). Gene Ontology (GO) results for genes showing CLAMP-dependent female SSS in embryos at the 0–2 hr/pre-MZT stage and for genes exhibiting CLAMP-dependent female and male SSS in embryos at the 2–4 hr/post-MZT stage, using ShinyGO v0.75c: Gene Ontology

*Figure 2 continued on next page*

*Figure 2 continued*

Enrichment Analysis with an FDR cutoff of 0.05. We compared the gene list with a background of all protein-coding genes in the *D. melanogaster* genome. The number of genes in each group (N) listed at the top of each dot plot. Number of genes involved in each GO category noted as the size of the circle and GO biological processes plotted according to degree of fold enrichment along the x-axis. The size of the circle increases as the number of genes in that category increases. The color of the circle represents significance (p-value, -log10FDR). GO categories for male embryos at the 0–2 hr/pre-MZT stage are not shown because the gene set is small and therefore no enriched GO categories were identified.

The online version of this article includes the following source data and figure supplement(s) for figure 2:

**Figure supplement 1.** Sex-specific differences in alternative splicing in early *D. melanogaster* embryos.

**Figure supplement 2.** CLAMP (chromatin-linked adapter for MSL proteins) inhibits aberrant alternative splicing in post-MZT (maternal to zygotic transition) male embryos.

**Figure supplement 3.** Validation of splicing differences at randomly chosen target genes where CLAMP (chromatin-linked adapter for MSL proteins) regulates sex-specific splicing by RT-PCR and qRT-PCR.

**Figure supplement 3—source data 1.** Source data for *Figure 2—figure supplement 3D*, *fus* transcript levels.

**Figure supplement 3—source data 2.** Source data for *Figure 2—figure supplement 3D, E, and F*, for *gapdh* transcript level.

**Figure supplement 3—source data 3.** Source data for *Figure 2—figure supplement 3E*, *wnd* transcript levels.

**Figure supplement 3—source data 4.** Source data for *Figure 2—figure supplement 3F*, *pep* transcript levels.

---

regulates its chromatin accessibility suggesting that we have identified relevant target genes (***Urban et al., 2017c***; ***Duan et al., 2021***; ***Gutierrez-Perez et al., 2019***). Furthermore, many of the validated target genes are themselves involved in splicing and chromatin regulation including those with known isoforms that specifically regulate AS such as *fus*, *pep*, and *sc35* (***Supplementary file 2***). In summary, loss of maternal CLAMP causes aberrant SSS of the majority of SSS zygotic transcripts including many that encode regulators of AS.

## Loss of zygotic CLAMP alters sex-specific AS during *Drosophila* development

Next, we asked whether in addition to maternal CLAMP, zygotic CLAMP influences SSS. Therefore, we analyzed total RNA-seq data from wild type control and *clamp²* null mutant (***Urban et al., 2017a***) third instar larvae (L3) and identified CLAMP-dependent SSS events (***Supplementary file 3a-c***). Similar to embryos, CLAMP-dependent AS events in male and female L3 larvae were largely sex-specific: **females** 139/189 (73.5%); **males** 161/211 (76.3%) (***Supplementary file 3a***, Columns H–J).

Zygotic CLAMP is also present in male and female cell lines derived from embryos. Furthermore, S2 and Kc cells are embryonically derived established models for male and female cells, respectively, that differ in their sex-chromosome complement (***Cherbas and Gong, 2014***; ***Cherbas et al., 1994***) and have been studied for decades in the context of dosage compensation (***Alekseyenko et al., 2008***; ***Straub et al., 2013***; ***Hamada et al., 2005***). Therefore, we also defined CLAMP-dependent splicing events by performing *clamp* RNAi in Kc and S2 cells. We first quantified all splicing events that differ between control populations of Kc and S2 cells using time2splice (***Supplementary file 4a***, ***Figure 3A***). Then, we identified CLAMP-dependent splicing events in cell lines (***Supplementary file 4b-d***). Because Kc and S2 cell lines are very similar except for their sex, we infer that these events are sex-specific, although it is possible that some of these events are sex-independent: (1) 45/46 CLAMP-dependent splicing events are female sex-specific in Kc cells; (2) 112/113 CLAMP-dependent splicing events are male sex-specific in S2 cells (***Supplementary file 4b***, Columns F–H).

Overall, there are fewer splicing events altered in the absence of CLAMP in cell lines compared with embryos consistent with cell lines remaining alive in the absence of CLAMP (***Soruco et al., 2013***) while embryos depleted for maternal CLAMP do not survive past ZGA and L3 *clamp* null larvae do not undergo pupation (***Duan et al., 2021***). In summary, there is zygotic CLAMP-dependent SSS in larvae and embryonically derived cell lines.

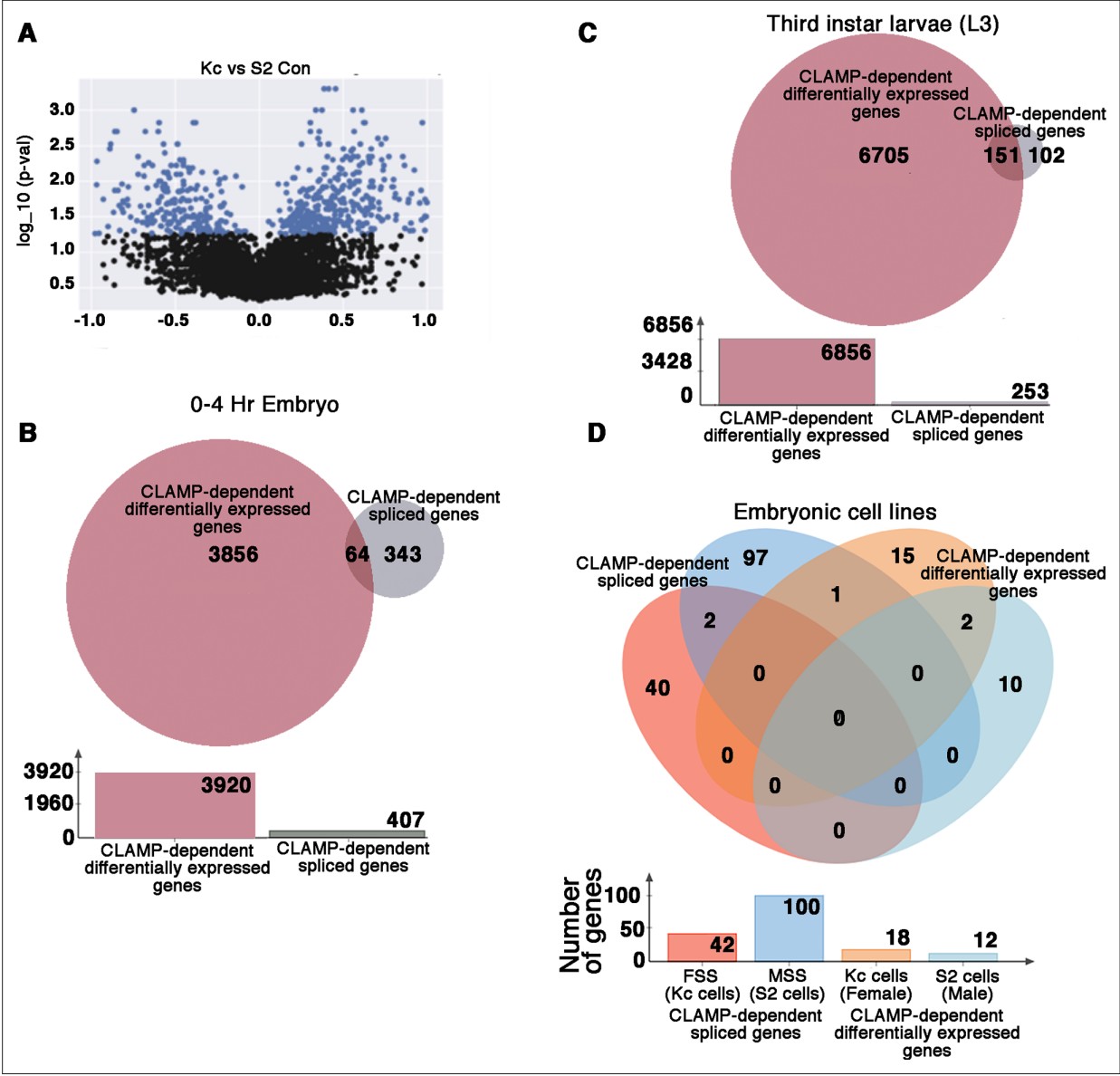

**Figure 3.** CLAMP (chromatin linked adapter for MSL proteins) has context-specific dual role in splicing and transcription at specific genomic loci. (**A**) Volcano plot showing $\log_{10}$_p-values for significant differences between percent spliced in (PSI) values for splicing events in female (Kc) and male (S2) *Drosophila* embryonic cell lines. Significantly changed splicing events (N=615) are labeled as blue dots (p<0.05 and PSI minimum ±0.2). (**B–C**) Venn diagram showing overlap between CLAMP-dependent spliced genes with CLAMP-dependent differentially expressed genes in third instar larvae (**A**) and 0–4 hr embryo (**B**). The total number of genes in each category is shown in the bar plot below the Venn diagram. (**D**). Venn diagram showing overlaps between dependent spliced genes in Kc (female) cells (pink circle) and S2 (male) cells (deep blue circle) with CLAMP-dependent differentially expressed genes in Kc (orange circle) and S2 cell lines (light blue circle). Bar plot shows the total number of genes in each category.

## Loss of CLAMP mis-regulates splicing and transcription at largely different sets of target genes in embryos and partially overlapping sets of targets genes in larvae

Because CLAMP regulates transcription in addition to splicing, we compared transcription and splicing target genes to each other and found that 84.3% (343/407) of genes which require CLAMP for their SSS in early embryos are not regulated by CLAMP at the transcription level as determined by comparing our differentially spliced genes with our published differential gene expression analysis (*Rieder et al., 2017*; *Figure 3B*). We also compared CLAMP-dependent differentially expressed genes (*Supplementary file 5a-b*) in third instar larvae with our list of CLAMP-dependent SSS genes

(*Supplementary file 3b-c*). In third instar larvae, 60% (151/253) of CLAMP-dependent SSS genes are also regulated at the level of transcription in contrast to 15.7% (64/407) of SSS genes in embryos (*Figure 3B and C*). Therefore, the genes that are regulated by CLAMP at the level of transcription and splicing are largely different in embryos and more similar in larvae.

Furthermore, in embryonic cell lines (Kc [female] and S2 [male] cells) almost all CLAMP-dependent spliced genes were regulated by CLAMP at the level of splicing and not transcription and many more genes are regulated at the level of splicing than transcription (*Figure 3D*). While 100 genes (112 splicing events) show CLAMP requirement for splicing in S2 cells, only 12 genes exhibit CLAMP-dependent differential gene expression. Similarly, in Kc cells, 42 genes (45 splicing events) show CLAMP requirement for splicing and only 18 genes show CLAMP-dependent expression (*Figure 3D*). Thus, the relative influence of the loss of CLAMP on splicing compared with transcription at target genes differs across different cellular contexts.

## CLAMP is highly enriched along gene bodies of genes at which SSS is mis-regulated by CLAMP depletion unlike genes at which transcription is mis-regulated

How does the loss of CLAMP alter SSS? We hypothesized that CLAMP would directly bind to DNA near the intron-exon boundaries of the genes where the loss of CLAMP alters splicing in contrast to genes where it regulates transcription. Therefore, we compared and contrasted the binding pattern of CLAMP at the CLAMP-dependent SSS genes and CLAMP-dependent transcriptionally regulated sex-biased genes in sexed embryos using CLAMP ChIP-seq data (#GSE133637). We found that the following percentages of all CLAMP-dependent SSS genes are bound by CLAMP across sexes and time points: 24.4% (29/119) in 0–2 hr female embryos, 8.3% (8/96) in 0–2 hr male embryos, 70.5% (146/207) in 2–4 hr female embryos, and 63.5% (66/104) in 2–4 hr male embryos (*Supplementary file 6a-b*). The increase in the percentage of genes bound by CLAMP in 2–4 hr embryos compared

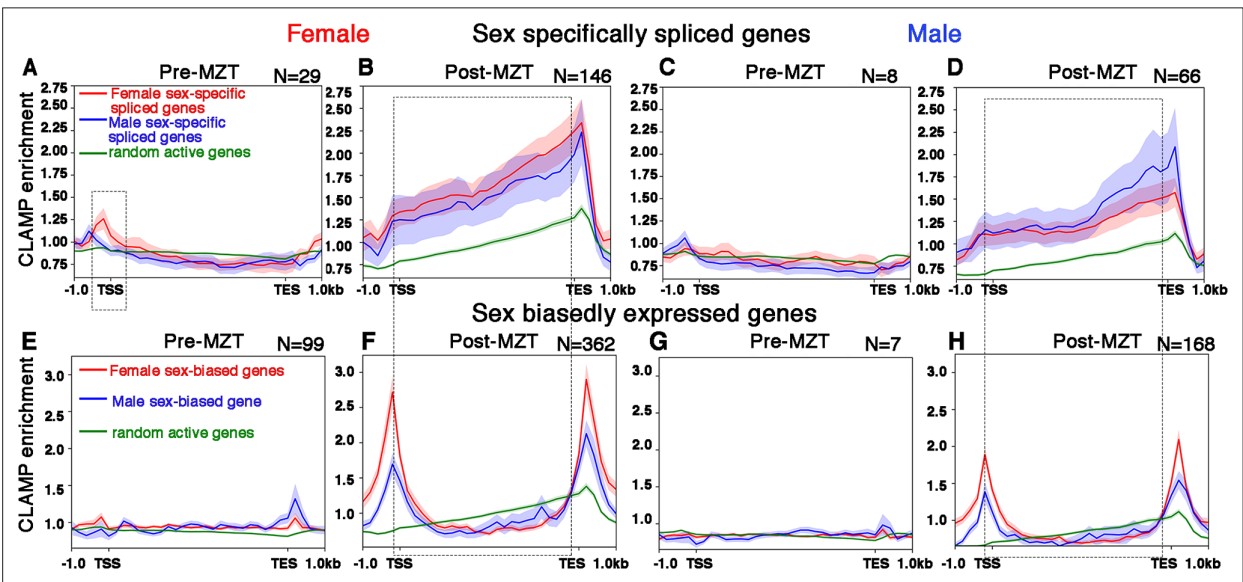

**Figure 4.** CLAMP (chromatin-linked adapter for MSL proteins) binds along the gene body of female and male sex-specifically spliced genes at the post-MZT (maternal to zygotic transition) embryonic stage. (**A–D**) Average profiles for CLAMP binding at pre-MZT and post-MZT embryonic stages in females (**A, C**) and males (**B, D**) for genes **spliced** female-specifically (red line) and male-specifically (blue line) during the pre-MZT (**A, B**) and post-MZT (**C, D**) stages. (**E–H**) Average profiles for CLAMP binding to genes **expressed** in a sex-biased manner in females (red line) and males (blue line) during pre-MZT (**E, F**) and post-MZT (**G, H**) stage. Green lines in (**A–H**) represent CLAMP binding at a random set of active genes used as a control (see Materials and methods for details). Stippled regions in (**A, C**) (female, 0–2 hr pre-MZT) denote chromatin around the transcription start site (TSS) with more CLAMP binding in female sex-specifically spliced genes vs. male sex-specifically spliced genes. The dotted boxes in (**A–H**) highlight the gene body regions in CLAMP-dependent sex-specifically spliced genes and genes with CLAMP-dependent sex-biased expression. Number of genes in each group denoted as N.

The online version of this article includes the following figure supplement(s) for figure 4:

**Figure supplement 1.** CLAMP (chromatin-linked adapter for MSL proteins) binds to chromatin near splice junctions.

with 0–2 hr embryos is consistent with the known increased number and occupancy level of CLAMP binding sites at the later time point (*Duan et al., 2021*).

Next, we generated average profiles for CLAMP occupancy at CLAMP-dependent SSS genes in females (**red lines**) and in males (**blue lines**) at 0–2 hr (pre-MZT) (*Figure 4A and C*) and 2–4 hr (post-MZT) (*Figure 4B and D*) in females (*Figure 4A and B*) and males (*Figure 4C and D*). We found that CLAMP occupies the gene bodies of many SSS genes that require CLAMP for their splicing.

Then, we compared the average CLAMP binding pattern at SSS genes (*Figure 4A–D*) to the CLAMP binding pattern at genes whose transcription but not splicing is both sex-biased and dependent on CLAMP (*Figure 4E–H*). In contrast to SSS genes where CLAMP occupies gene bodies, at genes that are expressed but not spliced in a CLAMP-dependent and sex-biased manner, CLAMP is enriched at the transcription start site (TSS) and transcription end site (TES) (area within the rectangular box is bounded by the TSS and TES in *Figure 4B, F, D, and H*). Furthermore, CLAMP binding is also modestly enriched at the TSS of female-biased expressed genes in females, consistent with enhanced CLAMP occupancy at the TSS of expressed genes (*Soruco et al., 2013*). As a control, we used a random set of active genes that are not regulated by CLAMP (**green lines** in *Figure 4A–H*) and we observed lower occupancy than at CLAMP-dependent genes. Overall, we found preferential binding of CLAMP along the gene bodies of genes that have CLAMP-dependent splicing in both females and males in contrast to TSS and TES binding at genes where transcription but not splicing requires CLAMP.

To determine whether the binding of CLAMP to gene bodies occurs close to splice junctions, we measured the distance between CLAMP peaks and the nearest splice junction (*Figure 4—figure supplement 1*). We found that CLAMP peaks are most frequently within 200–400 bp of either the start or the end of a splice junction, especially in SSS genes. The resolution of these measurements is limited by sonication and therefore it is possible that binding occurs even closer to splice junctions. We also found that CLAMP binds to chromatin closer to splice junctions at CLAMP-dependent SSS genes compared to genes with CLAMP-dependent sex-biased transcription in 2–4 hr female embryo samples which have the most target genes and CLAMP binding events which improves statistical robustness. The results were similar for all CLAMP peaks at SSS genes (*Figure 4—figure supplement 1C*) compared to peaks only present in introns (*Figure 4—figure supplement 1G*). Therefore, the binding pattern of CLAMP at splicing target genes is consistent with the loss of CLAMP altering SSS and is different from the binding pattern at genes where CLAMP regulates transcription.

## The loss of CLAMP alters splicing of sex determination pathway component genes

Next, we asked whether the loss of CLAMP regulates known key regulators of SSS. In *Drosophila*, sex-specific AS is regulated by the sex determination pathway. Sex-lethal (Sxl) is the master regulator of sex determination (*Salz and Erickson, 2010*) which drives subsequent SSS in females (*Bell et al., 1991*) of downstream effector genes giving rise to female-specific effector proteins that regulate female-specific splicing. Functional Sxl protein is only produced in females (*Salz and Erickson, 2010*; *Moschall et al., 2019*) because exon 3 in the *sxl* transcript contains a premature stop codon which is spliced out in females but retained in males (*Haussmann et al., 2016*). The absence of functional Sxl protein in males results in formation of male-specific effector proteins that regulate male-specific splicing. Therefore, we asked whether the loss of CLAMP affects AS of the *sxl* transcript.

To test whether CLAMP regulates *sxl* AS, we designed an RT-PCR assay to distinguish between the female-specific (excluding exon 3) and male-specific (including exon 3) versions of the *sxl* transcript (*Figure 5A*). To determine whether maternal CLAMP regulates splicing of the *sxl* transcript, we performed RT-PCR analysis of *sxl* splicing. In contrast to the much later larval stage (*Figure 5B*), in embryos, the male and female isoforms of Sxl have not become fully specified, consistent with the known autoregulation of *sxl* that occurs in embryos (*Salz and Erickson, 2010*; *Moschall et al., 2019*; *Urdaneta et al., 2019*; *Horabin and Schedl, 1996*). Despite the lack of complete specification of male and female *sxl* transcripts, our data show that loss of maternal CLAMP alters the SSS of *sxl* transcripts in 0–2 and 2–4 hr embryos because the male-specific transcript is not expressed in maternal CLAMP-depleted male embryos but is expressed in CLAMP-depleted female embryos (*Figure 5A*).

Next, we assayed the function of zygotic CLAMP in *sxl* splicing in three previously described fly lines: (1) our recessive *clamp* null mutant *clamp²* line (*Urban et al., 2017a*); (2) the heterozygous

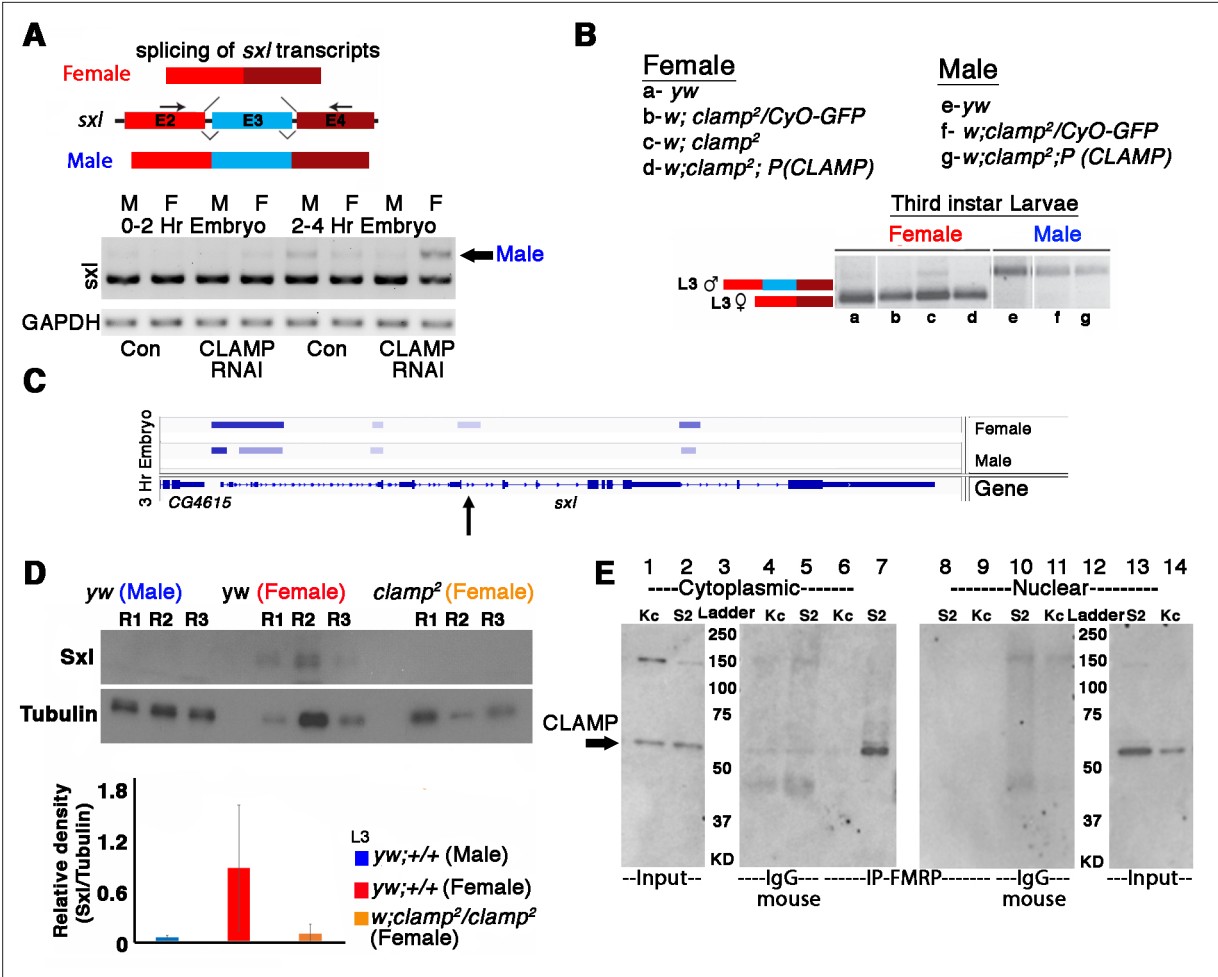

**Figure 5.** Alternative splicing of *sxl* transcript and Sxl protein levels is modulated by CLAMP (chromatin-linked adapter for MSL proteins) in females. (**A**) RT-PCR electrophoresis gel images (inverted colors) showing splicing of *sxl* transcripts in 0–2 and 2–4 hr sexed embryos in the presence and absence of maternal CLAMP with a representative schematic of the splicing event at the top of the gel image. The arrow indicates the male-specific *sxl* transcript (**number of replicates = 2**). (**B**) Electrophoresis gel image (inverted colors) showing splicing of *sxl* transcripts in third instar larvae of females and males of genotypes listed in the key (a–g) with a representative schematic at the top of the gel image. (**C**) IGV browser image showing CLAMP ChIP-seq peaks (rectangular boxes in light blue) at the genomic locus for the *sxl* gene in male and female 3 hr embryos. For each sample, the narrow peak file is shown which is generated after peak calling. Arrow in C indicates region of *sxl* gene where CLAMP differentially bind in females and not in males. (**D**) Western blot showing the level of Sxl protein in genotypes (three replicates for each) mentioned below each lane. Tubulin levels were used as a protein loading control. Below the blot is the relative quantification of Sxl protein levels compared with Tubulin and each genotype is represented by separately colored bars. (**E**) Western blot for CLAMP in cytoplasmic and nuclear protein fractions from Kc (female) and S2 (male) cells after IP (immunoprecipitation) using mouse anti-FMRP. IgG-mouse was used as negative control (lanes 4, 5 and lanes 11 and 12).

The online version of this article includes the following source data for figure 5:

**Source data 1.** Source data for *Figure 5A*, *sxl* and *gapdh* transcript levels.

**Source data 2.** Source data for *Figure 5B*, *sxl* transcript level.

**Source data 3.** Source data for *Figure 5D*, Sxl protein level.

**Source data 4.** Source data for *Figure 5D*, Tubulin protein level.

**Source data 5.** Source data for *Figure 5E*, western blot for detecting CLAMP (chromatin-linked adapter for MSL proteins) in IP-FMRP samples in cytoplasmic cellular fractions (lanes 1–7).

**Source data 6.** Source data for *Figure 5E*, western blot for detecting CLAMP (chromatin-linked adapter for MSL proteins) in IP-FMRP samples in cytoplasmic cellular fractions (lanes 8–14).

mutant *clamp²/CyO-GFP* line; (3) our rescue line which is homozygous for the *clamp²* allele and contains a rescue construct which is an insertion of wild type *CLAMP* gene at an ectopic genomic location. We measured CLAMP-dependent changes in AS of *sxl* and found that in homozygous *clamp²* female animals, there is a small but detectable amount of the longer male-specific *sxl* transcript (*Figure 5B*, lane c). This mis-regulation of *sxl* splicing is rescued by our *CLAMP*-containing rescue construct (*Figure 5B*, lane d). Furthermore, our L3 RNA-seq data demonstrate that CLAMP affects *sxl* splicing in females (*Supplementary file 3b*) and not *sxl* transcription (*Supplementary file 5b*). Consistent with the ChIP-seq binding pattern of CLAMP at the *sxl* locus on chromatin (*Figures 3 and 5C*) which shows enhanced binding in 2–4 hr embryos compared with 0–2 hr embryos, the loss of CLAMP alters splicing more strongly at the 2–4 hr time point compared with the 0–2 hr time point.

To test whether defects in *sxl* splicing altered Sxl protein levels, we performed western blots to quantify Sxl protein in wild type females and males and *clamp²* null females (*Figure 5D*). We observed a reduction in Sxl protein levels in females in the *clamp²* null background when compared with controls. Also, homozygous *clamp²* mutant males die before the late third instar larval stage likely due to loss of dosage compensation, and therefore it was not possible to measure the splicing of transcripts in male *clamp²* mutant larvae.

When comparing our RT-PCR assay measuring *sxl* splicing (*Figure 5B*) with western blotting analysis measuring Sxl protein levels (*Figure 5D*), we observed a more dramatic reduction in Sxl protein levels compared to changes in splicing. Regulation of translation by 5'UTR binding is a common mechanism for regulating protein production and stability (*Medenbach et al., 2011*; *Wilkie et al., 2003*). Therefore, we speculate that CLAMP may function in translational regulation of the Sxl protein because CLAMP interacts sex-specifically with the translation factor FMRP in the male but not the female cytoplasm (*Figure 5E*). Therefore, CLAMP might also have a distinct differential influence on translation in male and females depending on interacting translational regulators. Together, these data suggest that it is possible that mis-regulation of translation amplifies the CLAMP-dependent mis-regulation of splicing to generate a larger decrease in Sxl protein levels in the absence of CLAMP. Future experiments are required to test this hypothesis and decipher the underlying mechanisms. Independent of a potential effect on translation mediated by sex-specific FMRP interaction, we determined that the loss of CLAMP mis-regulates female-specific splicing of the *sxl* transcript.

Also, 88/407 (21.6%) of the CLAMP-dependent SSS genes are Sxl targets (*Supplementary file 7a*, #GSE98187). Therefore, we examined the splicing of other components of the sex determination pathway downstream of Sxl (*Figure 6A*). In embryos which lack maternal CLAMP (*Figure 6B*, lanes 2–3), the *dsx* female-specific transcript is aberrantly produced in males (*Figure 6B*, lanes 2–5). In contrast, the male-specific *dsx* transcript is not expressed in male embryos which lack CLAMP, similar to wild type female embryos (*Figure 6B*, lanes 7–10). We also observed male-specific *dsx* transcripts in female *clamp²* mutant larvae (*Figure 6C*, lane c). Therefore, *dsx* splicing is altered by the loss of maternal and zygotic CLAMP and CLAMP binds directly to the *dsx* gene locus (*Figure 6D*). These data suggest that the loss of CLAMP may regulate *dsx* splicing via both Sxl-dependent and Sxl-independent mechanisms.

In addition, we found that the loss of maternal and zygotic CLAMP mis-regulates splicing of the male-specific lethal-2 (*msl-2*) transcript (*Figure 6A*), which is male-specific because Sxl regulates its splicing, transcript stability, mRNA export, and translation in females (*Graindorge et al., 2013*; *Figure 6E and F*, lane c). To determine whether splicing defects are correlated with dysregulation of MSL-2 protein expression and localization to chromatin, we performed polytene immunostaining from female *clamp²* mutant salivary glands. In the absence of CLAMP, ectopic MSL2 protein (in red) is present at several locations on female chromatin in contrast to heterozygous females (*clamp²/CyO-GFP*) where the male-specific MSL-2 protein is not present on female chromatin as expected (*Figure 6G*). Similar to *dsx*, the *msl-2* gene is also bound by CLAMP (*Figure 6H*) and regulated by Sxl and, therefore, could be regulated through both direct and indirect mechanisms. Together, these data suggest that the loss of CLAMP affects the splicing of multiple components of the sex determination pathway.

## Discussion

Here, we define sex-specific alternatively spliced isoforms in pre- and post-MZT *D. melanogaster* female and male embryos genome-wide for the first time. We show that sex-specific transcript

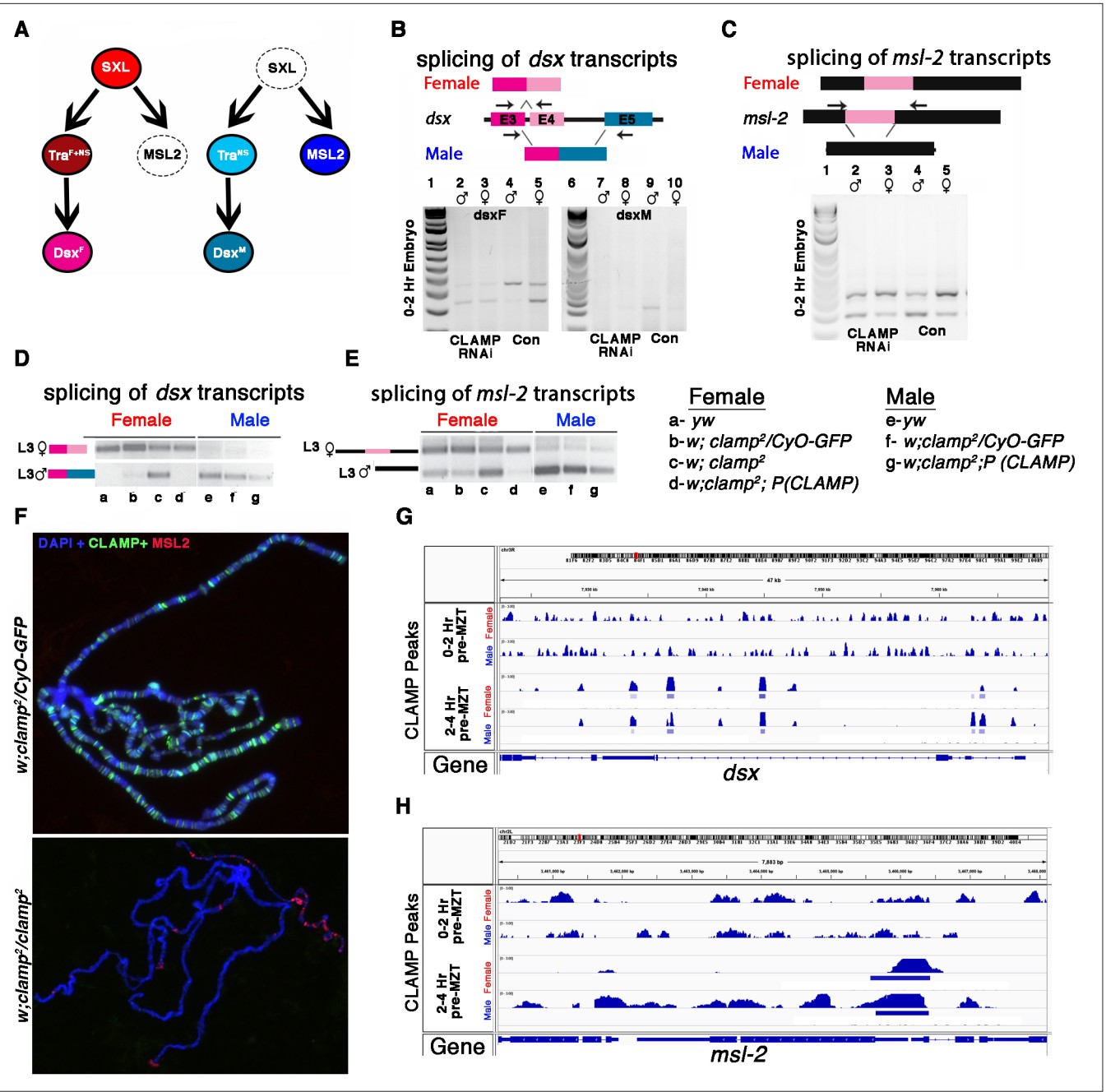

**Figure 6.** CLAMP (chromatin-linked adapter for MSL proteins)-dependent alternative splicing of components of the sex determination pathway. (**A**) The sex determination pathway in *Drosophila* is regulated by master regulator sex lethal (SXL). (**B–C**) RT-PCR electrophoresis gel images from 0 to 2 hr embryonic RNA samples (lanes 2–5 and 7–10) showing splicing of *dsx* (**B**) and *msl2* (**C**) transcripts in females (lanes 3, 5, 8, 10) and males (lanes 2, 4, 7, 9). Embryos were laid by *MTD-GAL4>GFP* RNAi control (lanes 4, 5, 9, 10) and *MTD-GAL4>CLAMP* RNAi (lanes 2, 3, 7, 8) females. The schematic above each gel image shows the female and male splice variants of the *dsx* (**B**) and *msl2* (**C**) transcripts. (**D**) IGV browser screen shot showing CLAMP peaks (rectangular boxes in light blue) at the genomic locus for the dsx genes in male and female 0–2 hr/pre-MZT (maternal to zygotic transition) and 2–4 hr/post-MZT embryos. The bigwig file (upper track) and the corresponding narrow peak file (lower track) are both shown. (**E–F**) Electrophoresis gel image from third instar larval samples (**a–g**) showing splicing of *dsx* (**E**) and *msl2* (**F**) transcripts in females (lanes a–d) and males (lanes e–g). a–g genotypes are the same as in panel (**A**). The schematics at the top of each gel image show female and male splice variants of *dsx* (**E**) and *msl2* (**F**) transcripts. (**G**) Fluorescent microscopy images of polytene chromosomes from the third instar salivary gland in the genotypes listed to the left of each panel (heterozygous control and *clamp²* null) show the distribution of CLAMP (green) and MSL2 (red) on chromatin (blue, DAPI). (**H**) IGV browser screen shot showing CLAMP peaks (rectangular boxes in light blue) at the genomic locus for the *msl-2* in male and female 0–2 hr/pre-MZT and 2–4 hr/post-MZT embryos. The bigwig file (upper track) and the corresponding narrow peak file (lower track) are both shown.

*Figure 6 continued on next page*

*Figure 6 continued*

The online version of this article includes the following source data for figure 6:

**Source data 1.** Source data for *Figure 6B*, *dsx* transcript level (lanes 1–5).

**Source data 2.** Source data for *Figure 6B*, *dsx* transcript level (lanes 6–10).

**Source data 3.** Source data for *Figure 6C*, *msl-2* transcript level (lanes 1–5).

**Source data 4.** Source data for *Figure 6D*, *dsx* transcript level in L3 female (lanes a–g, top row).

**Source data 5.** Source data for *Figure 6D*, *dsx* transcript level in L3 male (lanes a–g, bottom row).

**Source data 6.** Source data for *Figure 6E*, *msl-2* transcript level (lanes a–g).

diversity occurs much earlier in development than previously thought by generating the earliest data that define sex-specific transcript diversity across species (*Figures 1 and 2*). Furthermore, we determine that loss of a maternally deposited pioneer TF, CLAMP, mis-regulates sex-specific transcript diversity in early embryos causing the formation of many aberrant isoforms. Prior work on sex-specific transcript diversity (*Lott et al., 2014*; *Telonis-Scott et al., 2009*; *Paris et al., 2015*; *Gibilisco et al., 2016*; *Alekseyenko et al., 2008*; *Hartmann et al., 2011*; *Ranz et al., 2003*; *Zhang et al., 2007*; *Sun et al., 2015*; *Arbeitman et al., 2004*) either examined sex-biased differences in gene expression only or sex-specific transcript diversity much later in development in adult gonads or brain. To overcome the challenge of sexing early embryos before ZGA, we used a meiotic drive system that generates sperm with either only X or only Y chromosomes (*Rieder et al., 2017*) and measured both transcription and sex-specific transcript diversity generated by AS.

Because the transcript variants in both males and females encode genes that are involved in developmental processes, sex-specific developmental distinctions may occur earlier than previously thought. We demonstrate that a fundamental developmental trajectory differs between males and females from the initial hours of their existence long before gonad formation. Such early sex-specific transcript diversity may provide insight into how developmental disorders that originate before gonad formation can exhibit variable penetrance between sexes.

Different splice variants are produced at different frequencies over time and between sexes. To date, we lacked pipelines to characterize how these isoforms change over time. Therefore, we developed time2splice, which identifies mechanisms to regulate temporal and sex-specific AS by combining RNA-seq and chromatin occupancy data from ChIP-seq or CUT&RUN experiments. Time2splice has three parts: (1) temporal splicing analysis based on the SUPPA algorithm, (2) temporal chromatin occupancy analysis, and (3) temporal multi-omics integration. The pipeline and analysis steps can be accessed at: Ray, 2023; https://github.com/ashleymaeconard/time2splice.

We defined groups of genes in both males and females that undergo AS in a manner that is dependent on maternally deposited and zygotically expressed CLAMP. In embryos, genes whose splicing is altered in the absence of CLAMP are largely independent from those where CLAMP regulates transcription. In contrast, later in development there is a stronger overlap between genes where splicing and transcription are mis-regulated suggesting that additional cofactors may control how linked the dependence of splicing and transcription are on CLAMP at a specific gene.

The key question is: How does CLAMP, a ubiquitously expressed pioneer TF, affect SSS? We speculate that there are several mechanisms by which CLAMP influences SSS. CLAMP binds directly to intronic regions of approximately half of the CLAMP-dependent SSS genes (*Figure 4* and *Supplementary file 6a-c*) suggesting a potential direct role in regulating their co-transcriptional splicing. Many of these putative direct target genes are key regulators of AS which may explain the presence of indirect targets (*Supplementary file 2*) which requires future investigation to identify which factors directly mediate these events.

Furthermore, CLAMP regulates chromatin as a pioneer TF (*Duan et al., 2021*; *Urban et al., 2017b*) and recent literature links chromatin and splicing (*Agirre et al., 2021*; *Petrova et al., 2021*) and provides strong evidence that increased chromatin accessibility contributes substantially to the retention of introns during AS (*Petrova et al., 2021*). In addition, splicing-associated chromatin signatures have recently been identified (*Agirre et al., 2021*). For example, closed chromatin marks have recently been linked to exon exclusion and open chromatin has been linked to exon inclusion (*Agirre et al., 2021*; *Petrova et al., 2021*). Since CLAMP regulates chromatin accessibility (*Duan et al., 2021*) and CLAMP ChIP-seq data from sexed embryos (*Figure 5C*) shows that CLAMP binds differentially to

the *sxl* gene in females compared to males, it is possible that CLAMP might affect splicing via modulating chromatin accessibility.

Also, our proteomic analysis (*Urban et al., 2017c*) shows that CLAMP is sex-specifically associated with multiple spliceosome complex components including Squid, which is known to regulate SSS (*Hartmann et al., 2011*) specifically in females. Thus, it is possible that differential association between CLAMP and RBP spliceosome complex components in males and females regulates SSS. We therefore speculate that CLAMP may recruit RBP spliceosome complex components to regulate splicing by altering the chromatin environment or/and directly binding to target RNA transcripts.

We also investigated the effect of CLAMP at the *sxl* locus which encodes the master regulator of sex determination (*Salz and Erickson, 2010*). CLAMP binds near the early promoter of the *sxl* gene (SxlPe) (*Figure 5C*) in females. Because CLAMP binding sites are present near the promoter region of the *sxl* gene, we hypothesize that CLAMP regulates chromatin at exon 3 from a distance, consistent with our recent findings suggesting that CLAMP can mediate long-range chromatin interactions (*Bag et al., 2019*; *Jordan and Larschan, 2020*) and regulates chromatin accessibility most strongly several kb from its binding sites (*Urban et al., 2017b*). In the absence of CLAMP, we observe a stronger reduction in Sxl protein levels in females compared with changes in female-specific splicing (*Figure 5B and C*) which suggests that the loss of CLAMP might influence both *sxl* transcript splicing and translation. In addition, we have identified a sex-specific association between CLAMP and the FMRP translation factor in the cytoplasm which may explain changes in Sxl protein stability. Also, at both embryonic and larval stages we noted that splicing of the Sxl targets *dsx* and *msl-2* is affected by the loss of CLAMP (*Figure 6B–C and E–F*). CLAMP directly binds to these genes which are part of the approximately 52.3% (213/407) of CLAMP-dependent SSS genes that are directly bound by CLAMP (*Figure 6D and H*).

Our results support a hypothesis that the loss of CLAMP influences SSS through multiple mechanisms: (1) CLAMP directly binds to DNA of a subset of targets including the *sxl* gene itself and other key regulators of AS, (2) CLAMP influences splicing of other targets indirectly by regulating *sxl* splicing and Sxl protein levels, and (3) lastly, we speculate that CLAMP perhaps may further influence splicing via other unknown mechanisms such as interactions with RNA and RNA binding proteins involved in splicing with which it associates (*Urban et al., 2017c*).

Overall, our data show that the loss of maternal and zygotic CLAMP affects sex-specific AS, and influences the sex determination pathway. We show for the first time that loss of a maternal factor influences SSS during early embryonic development, highlighting how the maternal environment influences transcript diversity in the zygote including both activation of the zygotic genome and the processing of zygotic RNA products. We also present time2splice, a new pipeline to uncover mechanisms which drive such spatial-temporal transcript diversity by integrating splicing and chromatin occupancy data. Determining the mechanism by which the loss of CLAMP regulates SSS is a key future direction.

## Materials and methods

**Key resources table**

| Reagent type (species) or resource | Designation | Source or reference | Identifiers | Additional information |
|---|---|---|---|---|
| Genetic reagent (*D. melanogaster*) | MTD-Gal4 | Bloomington *Drosophila* Stock Center | BDSC:31777; FLYB:FBtp0001612; RRID:BDSC_31777 | FlyBase symbol: P{GAL4-nos.NGT} |
| Genetic reagent (*D. melanogaster*) | *UAS-CLAMPRNAi[val22]* | Bloomington *Drosophila* Stock Center | BDSC: #57008; | |
| Genetic reagent (*D. melanogaster*) | B[s]/Dp(2:y)CB25–4, y+, Rsp[s]B[s]; SPSD/CyO | Bloomington *Drosophila* Stock Center | BDSC: #64332; | |
| Genetic reagent (*D. melanogaster*) | +; SD72/CyO | Cynthia Staber, Stowers Institute | | |
| Genetic reagent (*D. melanogaster*) | 19–3, yw, Rsp[s]- | Cynthia Staber, Stowers Institute | | |

*Continued on next page*

*Continued*

| Reagent type (species) or resource | Designation | Source or reference | Identifiers | Additional information |
|---|---|---|---|---|
| Genetic reagent (*D. melanogaster*) | $y^1, w^{1118}; clamp^2 /CyO$ | *Urban et al., 2017a* | | |
| Cell line (*D. melanogaster*) | S2 | *Drosophila Genomic Resource Center (DGRC)* | FBtc0000181 | Cell line maintained in N. Perrimon lab; FlyBase symbol: S2-DRSC |
| Cell line (*D. melanogaster*) | Kc167 | *Drosophila Genomic Resource Center (DGRC)* | FBtc0000001 | |
| Antibody | Anti-CLAMP (Rabbit polyclonal) | Erica Larschan | RRID: AB_2195548 | IF (1:1000), WB (1:1000) |
| Antibody | Anti-Tubulin (Rabbit monoclonal) | Abcam | Cat# ab52866, RRID: AB_869989 | WB (1:5000) |
| Antibody | Anti-SXL (Mouse monoclonal) | Fatima Gebauer | | WB (1:500) |
| Antibody | Anti-MSL2 (Rat monoclonal) | Peter Becker | | IF (1:500) |
| Antibody | Anti-FMRP (Mouse monoclonal) | DSHB | 5B6 RRID:AB_528253 | IP-1:10 |
| Others | Anti-mouse IgG M-280 Dynabeads | Invitrogen, USA | Catalog number: 11202D | Magnetic Beads for IP |
| Others | Cell line RNA-seq | NCBI, GEO (This paper) | #GSE220439 | RNA sequencing data |
| Others | Third instar larvae (L3) RNA-seq | NCBI, GEO (This paper) | #GSE220455 | RNA sequencing data |
| Software, algorithms | SUPPA | *Trincado et al., 2018* | | |
| Software, algorithms | time2splice | This paper | https://github.com/ashleymaeconard/time2splice | |
| Software, algorithms | deeptools | *Ramírez et al., 2014* | http://deeptools.ie-freiburg.mpg.de | |

## Fly stocks and husbandry

*D. melanogaster* fly stocks were maintained at 24°C on standard corn flour sucrose media. Fly strains used: *MTD-GAL4* (Bloomington, #31777), *UAS-CLAMPRNAi[val22]* (Bloomington, #57008), Meiotic drive fly stocks +; SD72/CyO and 19–3, yw, Rsp[s]-B[s]/Dp(2:y)CB25-4, y+, Rsp[s]B[s]; SPSD/CyO (Bloomington, #64332) (both gifts from Cynthia Staber). These were crossed to obtained male and female embryo of desired genotypes according to *Rieder et al., 2017*.

## Cell culture

Kc and S2 cells were maintained at 25°C in Schneider's media supplemented with 10% fetal bovine serum and 1.4× Antibiotic-Antimycotic (Thermo Fisher Scientific, USA). Cells were passaged every 3 days to maintain an appropriate cell density. Both Schneider's Line S2 FBtc0000181 (S2-DRSC) and Kc (Kc167) FBtc0000001 cell lines used are of *D. melanogaster* embryonic origin from *Drosophila* Genomic resource center (DGRC). Both modENCODE line authenticated with no mycoplasma contamination and not from the list of commonly misidentified cell lines.

## Sample collection and western blotting

Salivary glands from third instar larvae were dissected in cold PBS and samples frozen in liquid nitrogen. Total protein from the samples was extracted by homogenizing tissue in the lysis buffer (50 mM Tris-HCl pH 8.0, 150 mM NaCl, 1% SDS, 0.5× protease inhibitor) using a small pestle. After a 5 min incubation at room temperature, cleared the samples by centrifuging at room temperature for 10 min at 14,000×*g*. To blot for CLAMP and Actin, 5 µg of total protein was run on a Novex 10%

Tris-Glycine precast gel (Life Technologies). To measure Sex-lethal protein levels, 20 µg of total protein was run on a Novex 12% Tris-Glycine precast gel (Life Technologies). Protein was transferred to PVDF membranes using the iBlot transfer system (Thermo Fisher Scientific) and probed the membranes for CLAMP (1:1000, SDIX), Tubulin (1:5000, Abcam), and SXL (1:500, a gift from Fatima Gebauer) antibodies using the Western Breeze kit following the manufacturer's protocol (Thermo Fisher Scientific). We quantified the relative expression of protein for SXL using the gel analysis tool in ImageJ software following the website's guidelines (*Schneider et al., 2012*). For each genotype, we first internally normalized the amount of SXL protein to Actin. Next, we determined the protein's relative expression by comparing the Tubulin normalized quantities to y[1], w[1118] female samples.

## Polytene chromosome squashes and immunostaining

Polytene chromosome squashes were prepared as previously described in *Rieder et al., 2017*. We stained polytene chromosomes with rabbit anti-CLAMP (1:1000, SDIX) and rat anti-MSL2 (1:500, gift from Peter Becker) antibodies. For detection, we used all Alexa Fluor secondary antibodies against rabbit and mouse at a concentration of 1:1000 and visualized slides at 40× on a Zeiss Axioimager M1 Epifluorescence upright microscope with the AxioVision version 4.8.2 software.

## Splicing assays for male- and female-specific transcripts

To test for the male and female splice forms of *sex-lethal, transformer, doublesex*, and *msl2*, total RNA was extracted from 10 third instar larvae from each genotype. We reverse-transcribed 2 µg of total RNA using the SuperScript VILO cDNA Synthesis Kit (Life Technologies) following the manufacturer's protocol. We amplified target sequences by PCR using primers designed to span alternatively spliced junctions. AS primer sequences for sxl FP-TGCAACTCACCTCATCATCC, sxl RP- GATGGCAGAGAA TGGGACAT, for tra FP- TGAAAATGGATGCCGACAG, tra RP- CTCTTTGGCGCAATCTTCTC, for dsx female transcript dsxFFP-CTATCCTTGGGAGCTGATGC, dsxF RP- TCGGGGCAAAGTAGTATTCG, for dsx male transcript dsxM FP- CAGACGCCAACATTGAAGAG, dsxM RP- CTGGAGTCGGTGGACA AATC, for msl2 FP- GTCACACTGGCTTCGCTCAG and msl2 RP- CCTGGGCTAGTTACCTGCAA were used.

## Validation of splicing results from time2Splice using qRT-PCR and RT-PCR assays

Total RNA was extracted from fifty 0–2 hr and 2–4 hr female and male embryos expressing *MTD-GAL4>GFPRNAi* (con) and *MTD-GAL4>CLAMPRNAi* (CLAMP depleted). Sexed embryos were obtained as described in *Rieder et al., 2017*. We reverse-transcribed 1 µg of total RNA using the SuperScript VILO cDNA Synthesis Kit (Life Technologies, USA) following the manufacturer's protocol. We amplified target sequences by PCR using primers designed to span alternatively spliced junctions (*Figure 2—figure supplement 3*) listed in *Supplementary file 7b* and Quick load Taq 2X Master mix (#M0271L, NEB, USA) according to the manufacturer's protocol (28 cycles). 10 µL of PCR product of each replicate for each gene was loaded in separate wells in 2% agarose gels and imaged using a ChemiDoc MP Imaging System (Bio-Rad, USA). All replicates for each gene were loaded on the same gel. The gel images were inverted and then quantified using the densitometry steps with the Fiji image analysis tool. qRT-PCR was carried out using 2X Azura Quant Green (#AZ-2120, Azura Genomics, USA) according to the manufacturer's instructions. Fold change between samples for each transcript was calculated the ΔCT method (*Schmittgen and Livak, 2008*). Student's t-tests were performed to determine significant difference between groups (two samples at a time). Three replicates for qRT-PCR samples and four replicates for RT-PCR samples were performed.

## Immunoprecipitation
### Nuclear and cytoplasmic extract preparation

Male (S2) and female (Kc) cells were grown to a cell concentration of $2\times10^6$ cells/mL in T25 tissue culture flasks. Cells were scraped from the flask, centrifuged for 5 min at 2500 rpm at 4°C. Supernatant was removed and cell pellets were washed twice in 5 mL of cold PBS. The washed cell pellets were then re-suspended in 5× volume of Buffer A (10 mM HEPES pH 7.9, 1.5 mM $MgCl_2$, 10 mM KCl, 0.5 mM DTT, 1× protease inhibitors). Cells were incubated on ice for 15 min before dounce homogenization with an A pestle. Cytoplasmic fraction was collected after centrifugation at 4°C for 20 min

at 700×*g*. The remaining nuclear pellet was re-suspended in three times volume in Buffer B (20 mM HEPES pH 7.9, 20% glycerol, 0.5% NP 40, 200 mM KCl, 0.5 mM EDTA, 1 mM EGTA, 1× protease inhibitors). Nuclei after re-suspension were dounce homogenized with a B pestle. Nuclear debris was then pelleted by centrifugation at 10,000×*g* for 10 min at 4°C. 1 mL aliquots of both cytoplasmic and nuclear fractions were prepared in 1.5 mL Protein LoBind Eppendorf tubes and flash-frozen in liquid nitrogen for storage at –80°C.

## Immunoprecipitation

Magnetic anti-CLAMP beads were prepared to a final concentration of 10 mg/mL by coupling rabbit anti-CLAMP antibody (SDIX) to magnetic beads, according to Dynabeads Antibody coupling kit (Thermo Fisher Scientific) instructions. Similarly, magnetic anti-FMRP beads were prepared using mouse anti-FMRP (5B6, DSHB, USA). Prepared anti-CLAMP, anti-FMRP, and purchased anti-IgG (anti-rabbit IgG M-280 and anti-mouse IgG M-280 Dynabeads raised in sheep, Invitrogen, USA) were blocked to reduce background the night before the immunoprecipitation. First, the beads were washed three times for 5 min in 500 L Tris-NaCl wash (50 mM Tris, 500 mM NaCl, 0.1% NP-40) by rotating at 4°C. The beads were next suspended in block buffer (3.3 mg/mL of yeast tRNA extract prepared in 20 mM HEPES, pH 7.9, 20% glycerol, 0.5% NP-40, 200 mM KCl, 1 mM EDTA, and 2 mM EGTA) and rotated overnight at 4°C. The next day, beads were washed three times for 5 min in the block buffer without yeast tRNA by rotating at 4°C. After the final wash, beads were re-suspended in the same amount of block buffer as the starting volume.

To 1 mL of previously prepared nuclear extract, 100 µL of blocked anti-CLAMP, anti-FMRP, or anti-IgG magnetic Dynabeads were added. The nuclear extracts/cytoplasmic extracts and beads were then rotated for 1 hr at 4°C. Afterward, the beads were collected and the supernatant discarded. The beads were then washed three times in Tris-NaCl wash (50 mM Tris, 500 mM NaCl, 0.1% NP-40) by rotating for 5 min at 4°C and cleared by using a magnetic rack. To elute proteins from the beads, 100 µL of 1% SDS was added, and the beads were boiled for 10 min at 95°C. To the eluate, 300 µL of ultrapure water was added, and the tubes gently vortexed. After collecting the beads on a magnetic rack, the eluate was saved in a clean Protein LoBind Eppendorf tube.

CLAMP was detected in IP-FMRP and IgG-mouse samples using rabbit anti-CLAMP (1:1000).

## RNA-sequencing

### RNA-seq in cell lines

15 µg each of *clamp* dsRNA and GFP dsRNA used for *clamp* RNAi and GFPRNAi (con), respectively, per T25 flask. Cells (Kc and S2) incubated with dsRNA in FBS minus media for 45 min and allowed to grow in media supplemented with 10% FBS for 6 days before harvesting. dsRNA targeting *gfp* (control) and *clamp* for RNAi have been previously validated and described (*Hamada et al., 2005*; *Larschan et al., 2012*). PCR products were used as a template to generate dsRNA using the T7 Megascript kit (Ambion, Inc, USA), followed by purification with the QIAGEN RNeasy kit (QIAGEN, USA). RNA was harvested using RNeasy mini plus kit (QIAGEN, USA). Two µg of total RNA was used for the construction of sequencing libraries. RNA libraries for RNA-seq were prepared using Illumina TruSeq V2 mRNA-Seq Library Prep Kit following the manufacturer's protocols. Hi-seq paired end 100 bp mRNA-seq performed. Data was submitted to the GEO repository (#GSE220439). For gene expression analysis, the DESeq2 pipeline was used. For identifying CLAMP-dependent splicing, our new time2plice pipeline was used.

### RNA-seq in third instar larvae (L3)

Total RNA was extracted from control (*yw*) and *clamp* mutant (*yw, clamp*[2]) male and female third instar larvae (three each) using Trizol (Invitrogen, USA). Messenger RNA was purified from total RNA using poly-T oligo-attached magnetic beads. After fragmentation, the first strand cDNA was synthesized using random hexamer primers followed by the second strand cDNA synthesis. The library was ready after end repair, A-tailing, adapter ligation, size selection, amplification, and purification followed by paired-end RNA-seq in Illumina Novaseq 6000. The sequencing data was run through a SUPPA-based time2splice pipeline to identify CLAMP-dependent SSS events. Data was submitted to the GEO repository (#GSE220455).

## Computational methods

### Time2splice tool

Time2splice is a new pipeline to identify temporal and sex-specific AS from multi-omics data that relies on the existing validated SUPPA method to identify differentially spliced isoforms (*Trincado et al., 2018*). This pipeline combines SUPPA with several additional scripts to identify SSS genes and sex-biased genes at different time points.

Importantly, these scripts are partitioned into separate script files to enable the user to use only the scripts that they need for their analysis. *Figure 1—figure supplement 1* describes the published methods and new scripts which we used in our analysis. Where boxes are numbered, the output from each step can be used as input for the subsequent step. Step D can be performed in any order depending on user needs. You can also see the README here: https://github.com/ashleymaeconard/time2splice for a detailed description of the methods.

### Tutorial section for time2splice

Preprocess (scripts/preprocess): Retrieve raw data, quality control, trimming, alignment. Perform steps as needed.

`1_parse_sraRunTable.sh`

Creates `time2splice/` folder structure, as well as `metadatafile.csv` and `SraAccList.txt` (which is needed for next command to get .fastq files).

`1_get_fastq_files.sh`

Retrieves.fastq files by passing in `SraAccList.txt` from aforementioned step.

`2_run_fastQC.sh`

Runs FastQC for all.fastq files in a given directory.

`3_run_trim_galore.sh`

Run Trim Galore! followed by FastQC to trim any reads below quality threshold.

`3_merge_lines.sh`

Merges all the different lanes of the same flow cell.fastq files.

`4_run_Bowtie2.sh`

or

`preprocess/4_run_BWA.sh`

or

`preprocess/4_run_HISAT2.sh`

Runs one or more of these three aligners (Bowtie2, BWA, or HISAT2) on.fastq data in a given directory.

```
5_plot_alignment.py
```

Plot the alignments from either one or two different aligners (Bowtie2 or HISAT2).

Temporal expression analysis (scripts/rna)

```
1_run_salmon.sh
```

Run salmon to quantify transcript expression for treatment and control samples. E.g.:

```
./1_run_salmon.sh /nbu/compbio/aconard/larschan_data/sexed_embryo/ /data/
compbio/aconard/splicing/results/salmon_results_ncbi_trans/ /data/compbio/
aconard/BDGP6/transcriptome_dir/pub/infphilo/hisat2/data/bdgp6_tran/genome.
fa 3 10 1 _001.fastq.gz
```

```
2_run_suppa.sh
```

Run SUPPA for treatment and control samples. E.g.:

```
./2_run_suppa.sh /data/compbio/aconard/splicing/results/salmon_results/ /
data/compbio/aconard/splicing/results/suppa_results_ncbi_trans/ /data/
compbio/aconard/BDGP6/transcriptome_dir/pub/infphilo/hisat2/data/bdgp6_
tran/genome.fa 20
```

```
3_suppa_formatting.py
```

Converts NM_ gene names to flybase name, then merging outputs from run_suppa (NM_ gene names by 1 TPM value column for each replicate)

```
4_suppa.sh
```

Identifies various forms of differential splicing (e.g. using PSI and DTU)

```
5_calc_total_alt_splicing_controls.py
```

Calculate and plot the proportions of AS (in pie chart) in control samples.

```
6_calc_total_alt_differential_splicing.py
```

Calculate and plot the proportions of AS (in pie chart) in treatment samples.

```
7_get_bias_genes.py
```

Retrieve male- and female-biased genes and create bed files for average profile plotting.

`8_plots_splicing.ipynb`

Plotting transcript expression using PSI and DTU measures.

`8_alt_plots_splicing.ipynb`

Alternative code base to plot transcript expression using PSI and DTU measures.

`9_plots_splicing_time.ipynb`

Plot AS genes within categories (all females, all males, females sex specific, male sex specific, female all rest, male all rest, female non-sex specific, male non-sex specific, female new sex specific, male new sex specific) over time.

## Temporal protein-DNA analysis (scripts/protein_dna)

`1_run_picard_markduplicates.sh`

Run Picard's MarkDuplicates in for all.sorted.bam files in a given directory.

`2_run_macs2.sh`

Runs MACS2 to call peaks for all.sorted.bam files in a given directory.

`3_run_macs2_fold_enrich.sh`

Generate signal track using MACS2 to profile transcription factor modification enrichment levels genome-wide.

## Temporal multi-omics integration (scripts/multio_analysis)

Note, there is no order to these scripts. Each analysis/results exploration is independent. More analysis scripts to come.

`overlap_protein_DNA_peaks.sh`

Runs Intervene to view intersection of each narrowpeak file.

`histogram_peak_val_intensity.ipynb`

Plot peak intensity for a given narrow peak file.

`get_coord_run_meme.sh`

Get coordinates of bed file and run through MEME.

```
alt_splicing_chi_squared.ipynb
```

Perform Chi-squared test on AS categories. MXE used in this example.

## Identification of sex-specifically splicing events

We quantified the amount of AS using an exon-centric approach to quantify individual splice junctions by measuring PSI for a particular exon using SUPPA within time2splice.

PSI = IR (included reads)/IR+ER (excluded reads)

The difference in PSI values (ΔPSI) between samples implies differential inclusion or exclusion of alternative exons among the two sample types. For example, a positive ΔPSI of 0.8 for an exon skip event means the exon is included in 80% of transcripts in the sample whereas a negative ΔPSI value implies reduced inclusion of the alternative exon. First, we determined significant differences in ΔPSI values for splicing events between the control female and male samples in 0–2 hr embryo (*Figure 2— figure supplement 1A*) and 2–4 hr embryo (*Figure 2—figure supplement 1D*) samples to identify CLAMP-independent SSS differences between males and females. We have included volcano plots to show how we defined significant differences with a p-value cutoff of p-value <0.05. Next, we determined the splicing events which are significantly affected by *clamp* RNAi in female and male samples (*Figure 2—figure supplement 1B–C, E–F*). Lastly, we compared the lists of CLAMP-independent to CLAMP-dependent SSS events identify the following categories of splicing events: (1) splicing events that differ between wild type males and wild type females and are also dependent on CLAMP; (2) CLAMP-dependent new SSS events: Splicing events that were not different when comparing wild type males and wild type females but do show sex-specific differences in the absence of CLAMP (*Figure 2B and C* and *Supplementary file 1a-h*).

## SSS event analysis

RNA-seq data from *Rieder et al., 2017* (#GSE102922), Kc and S2 cell line, and third instar larval data generated by us were analyzed using time2splice to determine sex-specifically splicing events. dmel-all-r6.29.gtf from BDGP6 in genomes (*DePristo et al., 2011*) was used to map each transcript identifier (ID) to gene ID and symbol, for .bed creation data for the associated chromosome, TSS and TES, and strand information were imported from Illumina (https://support.illumina.com/sequencing/sequencing_software/igenome.html). From the raw data after quality control, that is, FastQC (*Andrews, 2010*), Salmon (*Patro et al., 2017*) was used to quantify transcript expression for treatment and control samples. Calculated transcripts per million (TPM) values from SUPPA (*Trincado et al., 2018*) were used for all four replicates of female and male controls at both time points (before and after MZT). Each sample was filtered to include transcripts where the mean value is less than or equal to 3 TPMs per gene. The number of transcripts included at various thresholds was plotted from 1 to 10 and the fraction of genes filtered out begins to plateau around threshold 3. The PSI transcripts between females and males were compared at both 0–2 hr (pre-MZT) and 2–4 hr (post-MZT); Kc and S2 cells; and third instar larval stage, L3 (p-value of 0.05), thereby resulting in ΔPSI values and p-values for each transcription in each experimental condition comparison. Given these resulting delta transcript PSI values, significantly alternatively splice genes (p-value 0.05) were found between females vs. males 0–2 hr (pre-MZT) controls to show which genes are normally SSS pre-MZT. The same process was followed at 2–4 hr (post-MZT), in cell lines and third instar larvae. To then determine the SSS genes, the female RNAi experiment compared with the control ΔPSI gave the number of total alternative spliced transcripts pre-MZT, then considering those that are not shared with males, and are only expressed in females normally, this defined our sex specifically spliced set of genes for females pre-MZT. This process was also performed for males pre-MZT, for post-MZT sample; for S2 and Kc cell lines and for female and male L3.

## GO analysis

GO analysis was performed using the ShinyGO v0.75c: Gene Ontology Enrichment Analysis with an FDR cutoff of 0.05. However, specifically, in time2splice's script enrichment analysis, we provide R tool Clusterprofiler (*Wu et al., 2021*) for GO analysis. r implements GO analysis given an input gene set as a .txt file with a new line delimiter between genes. Given this input, it is converted to a vector of genes. The enrich GO function will return the enrichment GO categories after FDR correction. The FDR correction used is Benjamini-Hochberg to account for the expected proportion of false positives

among the variables (i.e. genes) for which we expect a difference. This was chosen over other methods such as the common Bonferroni method, as the Bonferroni correction controls the familywise error rate, where we are interested to account for false discoveries. The actual over-representation test itself is implemented in enrich GO according to *Yu et al., 2015*, where they calculate a p-value using the hypergeometric distribution (*Boyle et al., 2004*) and then perform multiple hypothesis correction. Importantly, while there are many tools to perform GO analysis, Cluster profiler was chosen due to its superior visuals and ability to handle multiple-omics types. This thus enables diverse additional analyses to be integrated into time2splice in the future such as ATAC-seq.

## ChIP-seq: data analysis

We used preprocessed ChIP-seq data from *Rieder et al., 2019* (#GSE133637), specifically the .bw and.broadPeak.gz files in our analysis using ChIPseeker (*Yu et al., 2015*) and deeptools (*Ramírez et al., 2014*). Specifically, when plotting the average profiles using deeptools, we achieved a baseline signal representing genome-wide binding taking into consideration the number of genes in other groups by the following procedure: of all genes that are on (no zero read-count genes), we sampled the number of the largest other group (to which we are comparing), and ran compute matrix on that subset. This process was repeated 500 times and the resulting 500 matrices were averaged to produce a representative signal. For motif analysis MEME (*Bailey et al., 2015*) suite was used.

## Acknowledgements

This work and funding to MR was supported by R35GM126994 to ENL from NIH. AMC is funded by the NSF Graduate Research Fellowship and CCMB, Brown University. We thank Bloomington stock center for fly lines.

## Additional information

### Funding

| Funder | Grant reference number | Author |
| --- | --- | --- |
| National Institute of General Medical Sciences | R35GM126994 | Mukulika Ray Erica Larschan |
| National Science Foundation | Graduate Research Fellowship | Ashley Mae Conard |

The funders had no role in study design, data collection and interpretation, or the decision to submit the work for publication.

### Author contributions

Mukulika Ray, Conceptualization, Data curation, Formal analysis, Validation, Investigation, Visualization, Methodology, Writing – original draft; Ashley Mae Conard, Conceptualization, Data curation, Software, Formal analysis, Visualization, Methodology, Writing – review and editing; Jennifer Urban, Validation, Investigation, Visualization, Methodology; Pranav Mahableshwarkar, Data curation, Software, Formal analysis, Methodology; Joseph Aguilera, Data curation, Software, Formal analysis; Annie Huang, Data curation, Software, Formal analysis, Investigation; Smriti Vaidyanathan, Data curation, Software; Erica Larschan, Conceptualization, Resources, Formal analysis, Supervision, Funding acquisition, Visualization, Project administration, Writing – review and editing

### Author ORCIDs

Mukulika Ray 
Jennifer Urban 
Erica Larschan 

### Decision letter and Author response

Decision letter https://doi.org/10.7554/eLife.87865.sa1
Author response https://doi.org/10.7554/eLife.87865.sa2

# Additional files

## Supplementary files
- Supplementary file 1. List of sex-specific splicing events in early Drosophila embryos. (a–h) List of CLAMP (chromatin linked adapter for MSL proteins)-dependent sex-specific splicing events in 0–2 hr pre-MZT (maternal to zygotic transition) embryo (a–d) and 2–4 post-MZT hr embryo (e–h).
- Supplementary file 2. Summarizing the results and functions of the validated target genes at which splicing requires CLAMP (chromatin linked adapter for MSL proteins).
- Supplementary file 3. List of sex-specific splicing events in third instar Larvae (L3). (a–c) List of all and sex-specific splicing events in *Drosophila* third instar larvae (L3) that require CLAMP (chromatin linked adapter for MSL proteins) (a), list of genes whose splicing in females (b) and males (c) is affected by loss of CLAMP.
- Supplementary file 4. List of splicing events in S2 and Kc Drosophila embryonic cell lines. (a–d) List of differential splicing events between Kc (female) and S2 (male) cell lines (a), all CLAMP (chromatin linked adapter for MSL proteins)-dependent splicing events in *Drosophila* sexed embryonic cell lines (b), female- (c) and male- (d) specific splicing events in *Drosophila* sexed embryonic cell lines affected by loss of CLAMP.
- Supplementary file 5. List of CLAMP dependent differentially expressed genes in third instar larvae (L3). (a–b) List of all CLAMP (chromatin linked adapter for MSL proteins)-dependent differentially expressed genes in *Drosophila* male (a) and female (b) third instar larvae (L3).
- Supplementary file 6. CLAMP DNA binding data listing sex-specifically regulated genes bound by CLAMP. (a–c) List of CLAMP (chromatin linked adapter for MSL proteins)-dependent spliced genes which are directly bound by CLAMP (a), list of sex-specifically spliced genes (b) and sex-biasedly expressed genes (c) in 0–2 and 2–4 hr male and female pre-MZT (maternal to zygotic transition) and post-MZT embryos, respectively.
- Supplementary file 7. List of Sxl targets which require CLAMP for sex-specific splicing. (a–b) List of CLAMP (chromatin linked adapter for MSL proteins)-dependent male and female specifically spliced genes which are direct SXL targets (a). List of primers used to validate splicing events requiring CLAMP in *Figure 2—figure supplement 3* (b).
- MDAR checklist

## Data availability
Sequencing data have been deposited in GEO under accession codes #GSE220455 and #GSE220439. All data generated or analyzed during this study are included in the manuscript and supporting file. Source data files have been provided for Figure 2—figure supplement 3, Figure 5, and Figure 6. Source data used to generate all the figures, graphs, and Venn diagrams are provided in *Supplementary files 1–7*.

The following datasets were generated:

| Author(s) | Year | Dataset title | Dataset URL | Database and Identifier |
|---|---|---|---|---|
| Aguilera J, Mahableshwarkar P, Ray M, Larschan E | 2023 | Effect of loss of transcription factor CLAMP on sex-specific splicing in *Drosophila* third instar larvae (L3) stage | https://www.ncbi.nlm.nih.gov/geo/query/acc.cgi?acc=GSE220455 | NCBI Gene Expression Omnibus, GSE220455 |
| Urban J, Mahableshwarkar P, Ray M, Larschan E | 2023 | Identifying splicing targets of CLAMP by mRNA-sequencing | https://www.ncbi.nlm.nih.gov/geo/query/acc.cgi?acc=GSE220439 | NCBI Gene Expression Omnibus, GSE220439 |

The following previously published datasets were used:

| Author(s) | Year | Dataset title | Dataset URL | Database and Identifier |
|---|---|---|---|---|
| Rieder LE, Koreski KP, Boltz KA, Kuzu G, Urban JA, Bowman S, Zeidman A, Jordan III WT, Tolstorukov MY, Marzluff WF, Duronio RJ, Larschan EN | 2017 | Histone locus regulation by the *Drosophila* dosage compensation adaptor protein CLAMP | https://www.ncbi.nlm.nih.gov/geo/query/acc.cgi?acc=GSE102922 | NCBI Gene Expression Omnibus, GSE102922 |
| Rieder L, Jordan W, Larschan E | 2019 | Targeting of the dosage-compensated male X-chromosome during early *Drosophila* development | https://www.ncbi.nlm.nih.gov/geo/query/acc.cgi?acc=GSE133637 | NCBI Gene Expression Omnibus, GSE133637 |
| Moschall R, Rossbach O, Lehmann G, Kullmann L, Eichner N, Strauss D, Strieder N, Meister G, Krahn M, Medenbach J | 2019 | Flipping the switch on Sex-lethal expression: Sister of Sex-lethal antagonizes Sxl-dependent alternative splicing to maintain a male-specific gene expression pattern in *Drosophila* (RIP-Seq) | https://www.ncbi.nlm.nih.gov/geo/query/acc.cgi?acc=GSE98187 | NCBI Gene Expression Omnibus, GSE98187 |

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
