## [Editor Report]

In this manuscript, the authors describe the earliest differences in sex-specific splicing in *Drosophila* embryos or any animal for that matter. Based on solid data, they report the important finding that differences arise already during the first few hours of embryogenesis and that a maternally-deposited pioneer transcription factor contributes to generating these differences. The authors also provide a bioinformatics pipeline to analyze splicing over time.

---

## [Decision Letter]

**Decision letter after peer review:**

[Editors’ note: the authors submitted for reconsideration following the decision after peer review. What follows is the decision letter after the first round of review.]

Thank you for submitting the paper "Sex-specific transcript diversity is regulated by a maternal pioneer factor in early *Drosophila* embryos" for consideration by *eLife*. Your article has been reviewed by 3 peer reviewers, one of whom is a member of our Board of Reviewing Editors, and the evaluation has been overseen by a Senior Editor. The reviewers have opted to remain anonymous.

Comments to the Authors:

We are sorry to say that, after consultation with the reviewers, we have decided that this work will not be considered further for publication by *eLife*.

The three reviewers agreed on a number of points, with the main ones being that:

1. the mechanistic conclusions on how CLAMP may affect alternative splicing are not fully substantiated by the data;

2. that the effect of CLAMP loss of function on Sxl is quite minimal – despite the effect at the protein level appearing to be very substantial; this apparent discrepancy raises several concerns about what direct vs. indirect effects of manipulating CLAMP.

The reviewers had several specific points in addition to this, which are attached below.

*Reviewer #1 (Recommendations for the authors):*

In this manuscript, the authors describe the earliest systematic differences in sex-specific splicing in *Drosophila* embryos or any animal for that matter. They find that differences arise already during the first few hours of embryogenesis and also identify a maternally-deposited pioneer transcription factor that contributes to generating these differences. The authors also provide a bioinformatics pipeline to analyze splicing over time.

The main strength of this paper is that the authors were able to generate pure populations of male or female embryos (using a recently published genetic system in *Drosophila*), and they exploited this by generating numerous genome-wide datasets. Their analyses revealed an interesting link between a maternally deposited transcription factor and alternative splicing, in particular, of genes that are differentially spliced between males and females.

A weakness of this paper is that several mechanistic conclusions are drawn from rather correlative experiments. So while the main observations are very interesting, the mechanistic model should be interpreted more cautiously.

This manuscript presents a lot of data exploring the question of regulation of sex-specific splicing by a maternally deposited factor, in early *Drosophila* embryos. These data show:

– Loss of maternal CLAMP affects alternative splicing of 200-400 transcripts, with a trend to affect a bit more the mutually exclusive exon category.

– This is a small fraction of the total alternative splicing events in embryos at these times, but a larger fraction of the sex-specific alternative splicing (30-60%).

– Loss of maternal CLAMP causes both loss and gain of alternative splicing events.

– ChIP-seq of CLAMP shows two types of distribution over genes: on genes whose level is affected by CLAMP, it is enriched at the ends of the genes; on genes whose alternative splicing is regulated by CLAMP, there is an enrichment over the whole gene body.

– iCLIP of CLAMP shows binding to RNA, predominantly on chromatin (experiment done in cell lines). It binds mostly mRNAs but also other RNAs

– CUT&RUN of the helicase MLE also has a dual type of enrichment over genes. Upon loss of maternal CLAMP, the signal in males looks globally lower, in females less so. But other peaks are also gained, suggesting a re-distribution.

– CLAMP is necessary for sex-specific splicing of Sxl in a manner that correlates with chromatin accessibility over the alternative exon.

The observations presented here are very interesting, but a few critical aspects of the mechanism are over-interpreted. For example, I find it hard to know whether the association with snoRNAs found in iCLIP is meaningful. By the same criteria, we should conclude that CLAMP associates with tRNAs. I think it's more likely that the proximity of spliceosomes to chromatin produces the observed signal. Also, the specific focus on particular peaks of CLAMP and MLE is difficult to interpret without more global views of how these very general proteins redistribute along the genome. Also, the connection between MLE redistribution and alternative splicing is not really explored adequately. I don't think the authors need more data for this paper, but I would suggest more concise and accurate descriptions of the data and more cautious interpretations.

Regarding the last experiments looking at splicing of Sxl, it seems unclear to me why the authors switched from the RNAi tool to ablate the maternal contribution, to a mutant allele which (if I understand correctly) ablates the zygotic contribution. It is also unclear why even though a small fraction of Sxl is mis-spliced in females, the effect on the protein level is so dramatic (!). This suggests that something else is happening in these embryos that lack CLAMP. This actually raises some doubt about what causes all the effects in the earlier part of the paper, but we cannot really compare these experiments given that the authors used very different genetic tools.

*Reviewer #2 (Recommendations for the authors):*

Ray, Conard et al. describe the role of the CLAMP transcription factor in the regulation of sex-specific alternative splicing in *Drosophila* embryos, larvae, and tissue culture cells. The results are presented from three main lines of investigation: (1) genomics in early embryos, including sex-specific RNA-seq, which is analyzed with "time2splice" a newly developed pipeline for detection of alternative splicing events; (2) iClip in cultured S2 and Kc cells. (3) Validation of CLAMP's effect on alternative splicing of core sex determination factors Sex Lethal, Transformer, and Doublesex. There is very much data included in this manuscript, and the presentation of the genomics data will require substantial clarification for readers accurately to interpret the results. At the heart of the manuscript is the observation that clamp mutants demonstrate aberrant expression or loss of expression of sex-specific genes, and the argument presented here is that this is largely due to the misregulation of splicing in clamp mutants. While the genomics data suggest that CLAMP is necessary for certain sex-specific alternative splicing events and that CLAMP interacts with splicing factors, CLAMP appears to only have a small effect on the specific examples of sex-dependent alternative splicing (Sxl, Tra, *Dsx*) presented as validation. The findings also catalog for the first time the splice variants present at the maternal-to-zygotic transition, but the current analysis of these data leaves open the question of whether such alternative splicing events are associated with zygotic transcripts, and whether the magnitude of CLAMP's effect in this process is significant.

– The title of the manuscript touts the regulation of alternative splicing by a maternal pioneer factor, but a substantial proportion of the data is derived from cultured cells or third instar larvae, where the maternal contribution of CLAMP is not substantial. It is also unclear what pioneering per se has to do with the mechanism the authors propose and what little connection they present is not focused on in any validation or mechanistic follow-up work. I recommend changing the title to de-emphasize CLAMP's maternal expression and pioneer activity.

– Throughout the manuscript, but particularly within the description of the RNA-seq results, the number of objects (genes, transcripts, splicing events, peaks, etc…) are rarely stated explicitly in the text, but are instead sometimes part of figures or legends, if they are stated at all. Please edit the manuscript throughout to indicate the number of objects being compared and include percentages of the relevant group when discussing specific categories (e.g., x% of total genes/transcripts are alternatively spliced (n/N). Y% (n) of alternatively spliced genes/transcripts are sex-specific. Of these z% (n) are zygotically expressed.). As written, I was unable to clearly evaluate the biological conclusions for the first half of the manuscript because I had to rely on non-quantitative descriptors provided by the authors to glean magnitudes: for example, "very low levels," (line 260).

– In terms of the magnitude of the effect: piecing together information in Figures 1 and 2, there appears to be 10891 total (genes? transcripts?) in the 0-2 hour female RNA-seq data. Figure 1B implies that 16.27% (1771) of these are alternatively spliced. Can the authors comment on what an expected range of alternative splicing would be for a 'typical somatic cell' of any sort?

– For Figure 2D and associated text, the authors address whether CLAMP-dependent sex-specific alternative splicing is observed mainly in zygotic genes. I have several issues here. Mainly, I am unclear on how this comparison is being made (partly because of the lack of numbers in the text). The numbers of sex-specific AS genes in the legend are different than the numbers in the Venn diagrams. From the minimal explanation of how this was done, the impression is given that if an AS gene was not one of the 841 maternal genes, then it is likely to be zygotic. This does not follow logically, given that most of the 0-2 hour transcripts (~10891?) will by definition be maternal, and that non-membership in this limited list of 841 maternal genes (from which source?) cannot directly imply that the gene is zygotic, since most of the remaining transcripts will be maternal but not included in the limited list of 841. There are near-exhaustive gene lists of purely zygotic genes (DeRenzis/Wieschaus), and maternal-zygotic genes (Chen/Zeitlinger, Kwasnieski/Bartel). This analysis may be enhanced by enumerating the fraction of purely zygotic genes and maternal-zygotic genes. This is an important analysis and should be (1) re-done, and (2) documented extensively (with accurate numbers).

– Corollary to the above point: the 10891 number (Figure 2A): does that refer to unique transcripts (tx) or unique genes (gn)? The enumeration requested in the above point should clearly state that the numbers are tx or gn, and if tx, the total number of gn represented by that value should be cited. Comparisons with published gene lists should be done in the appropriate 'unit' (tx or gn), dictated by the published list.

– The authors should comment on how a maternally supplied transcript could be alternatively spliced in a sex-specific manner. At least some of the sex-specific alternatively spliced genes are identified as being maternal in Figure 2. Are these maternal genes that are also zygotically expressed? Are the maternal isoforms consistent with the female zygotic isoform? Any detection of sex-dependent alternative splicing in solely maternal genes (i.e., not zygotically expressed) could indicate issues with the computational approach for scoring AS events and should be discussed.

– One of the differences between Kc and S2 cells is their sex, but this does not mean that any difference observed between the two cell types is sex-specific. The section beginning at line 335 reads as if 100% of the differences between Kc and S2 cells is interpreted as a sex-specific difference. In any case, I also felt that the results from this section should be confirmed in embryos somehow, perhaps through an RNA IP experiment. snRNAs should be abundant enough that they can be detected in a CLAMP IP, even from early male or female embryos.

– Section beginning line 514: In general, one of the weaknesses of this paper is that it switches between embryos, larvae, and cultured cells, without much critical evaluation of whether such different contexts impact the strength of the conclusions. In this case, I am puzzled why the authors choose to solely rely on MNase-seq in cultured cells to make a point about CLAMP-dependent chromatin accessibility when they have also performed ATAC-seq on CLAMP-knockdown embryos. The observation that loss of CLAMP leads to a greater amount of accessibility at Sxl exon 3 specifically in Kc cells (which is presumed to be because these are female cells but could instead be a cell-type specific effect independent of sex). Such a large effect should be evident in mixed-sex embryo collections from a CLAMP knockdown, and these data should be shown and presented in the Results. Are sex-specific differences in CLAMP binding observed by ChIP-seq at this locus? This data would be essential to show as well.

– The magnitude of the effect of CLAMP loss of function on Sxl splicing, however, does not seem to be very large, given the near absence of Sxl protein in female larvae. Can the authors clarify how they interpret this discrepancy? The same could be said for the MXL results. Is the regulation of splicing only a minor function of CLAMP?

– I have not yet reviewed the supplemental tables for completeness and suitability for use in follow-up studies. This should be revisited during the consultation.

*Reviewer #3 (Recommendations for the authors):*

In flies, it is well established that sex determination is controlled by gender-specific alternative splicing of the sxl gene. The current study extends the catalog of embryonic alternative splicing events, including numerous new gender-specific splicing events. The primary data set is RNA-seq from pre- and post-zygotic gene activation gender-specific embryo samples collected using the meiotic drive. The study identifies 92 transcripts differentially spliced between genders at 0-2 hours post fertilization, and 138 at 2-4 hours post-fertilization. A small subset (4) of splicing events were validated by alternative methods. In general, these data are convincing, though more validation would strengthen confidence in the data set.

The data are then compared to existing RNA-seq where maternal CLAMP is depleted. CLAMP is a candidate maternally deposited alternative splicing regulator. Between 30-50% of gender-specific splicing events are CLAMP-dependent, while only 2-3% of total splicing events are CLAMP-dependent. The authors suggest these results indicate a specific role for CLAMP in regulating gender-specific alternative splicing. The overall number of CLAMP- and gender-specific alternative splicing events is fairly low (<50) compared to the total number of alternative splicing events detected (>10,000). Having said that, the enrichment is significant by Fisher's Exact Test, and subsequent binding experiments provide additional support for the model.

Next, the authors performed gender-specific ChIP-seq to map positions of CLAMP binding in embryos as a function of the developmental stage. Approximately half of the CLAMP-dependent gender-specific alternative splicing events show CLAMP enrichment on the DNA of alternatively spliced genes. This suggests (but does not prove) that CLAMP could be directly regulating co-transcriptional alternative splicing of gender-specific events. CLAMP also binds to some gender-specific mRNAs revealed by iCLIP data sets from male and female cell lines. This binding is enriched in RNAs that co-fractionate with chromatin, suggesting that DNA and RNA precipitation is coupled. CLAMP also appears to bind to snRNAs and components of the spliceosome, and loss of maternal CLAMP causes redistribution of the MLE complex, especially in male samples.

Finally, the authors show that CLAMP plays a role in regulating sxl alternative splicing, and this role correlates to CLAMP-dependent chromatin formation.

All told, the experiments point to a model where CLAMP regulates alternative splicing for a limited subset of embryonic transcripts, about half of which are gender-specific alternative splicing events, through a mechanism that involves DNA binding, RNA binding, and chromatin conformation. This is a fascinating outcome for a variety of reasons, as the concept of broad parental control over progeny splicing patterns has not been widely explored. It seems clear that the mechanism proposed accounts for a small fraction of the observed embryonic alternative splicing. There is no evidence that the novel alternative splicing events detected are important for embryogenesis or sex determination. Nevertheless, the results do move the field forward and provide testable hypotheses that can be addressed in future studies.

In all cases where data is presented as a percentage, it would be MUCH CLEARER if both the numerator and denominator were presented, along with a p-value. For example, on line 279, it states "43.8% of all CLAMP-dependent sex-specifically spliced genes are bound by CLAMP:" This should be followed with (num/denom, p-value XXX) so it is clear to the reader that this percentage is meaningful and how large (or small) of a list of genes it describes. This should be done throughout the manuscript.

Several key experiments are relegated to supplementary information (for example, the volcano plots in figure S2). Where possible, critical experimental data should be moved into the body of the manuscript and confirmatory analysis placed in the supplement.

Some of the rationale was missing from the text, for example, the chromatin fractionation for the iCLIP data sets. I think I understand why this was done, but it should be presented.

The paper would be stronger with functional studies to assess the biological importance of the CLAMP-dependent sex-specifically spliced genes, although the manuscript is already overloaded with experiments and it strikes me as unreasonable to request more.

The work as presented is difficult for the reader to get through. The paper would benefit from significant rewriting. I found significant overlap between the introduction and Results section, with several concepts and rationale presented in both sections. Also, in one case (concerning sxl) introductory material was first presented in the Results section. The reader would benefit from a more succinct presentation. In fact, the authors might wish to consider splitting the work into more digestible chunks, for example, a description of gender-specific alternative splicing events/stage-specific splicing events along with a more detailed description of the pipeline used to identify them, and a CLAMP paper with more functional characterization of the impact of CLAMP targets on embryogenesis. Ultimately, this decision is up to the authors, but I would ask them to consider it for readability/clarity's sake.

There are some typos that should be corrected (line 344 "most CLAMP RNA binds to hundreds of RNA", line 340 "Although CLAMP do not have a canonical RNA recognition motifs").

[Editors’ note: further revisions were suggested prior to acceptance, as described below.]

Thank you for resubmitting your work entitled "Sex-specific transcript diversity is regulated by a maternal transcription factor in early *Drosophila* embryos" for further consideration by *eLife*. Your revised article has been evaluated by Kevin Struhl (Senior Editor) and a Reviewing Editor.

The manuscript has been improved but there are some remaining issues that need to be addressed, as outlined below:

After careful review and much further discussion, the reviewers agree that the premise is very interesting and the data are original and valuable. However, they also all agree that the paper tries to tell a story (or even multiple stories) that are not fully substantiated by the data. The data are overinterpreted in a number of places and that leads to a main storyline of CLAMP being a critical regulator of splicing that the reviewers (4 in total by now) do not think is substantiated. They do not dispute that there are effects on splicing, but they do not think that the data support the mechanism proposed to explain these, or that these changes are functionally meaningful (relative to other functions of CLAMP).

The editors and reviewers would be willing to evaluate a revised version of this manuscript, but this would require a very significant rewrite to more accurately and clearly present and interpret the data. No new experiments are required (although they can be added). At this point, the reviewers are undecided about whether the current version should be rejected or sent back for revision. To facilitate the decision and to save time, they have suggested that you first send back a revised abstract that indicates how you will address the main criticism. If this revised abstract satisfactorily addresses the main issue, the decision will be "revise" under the assumption that the bulk of the paper is changed in accord with the revised abstract.

*Reviewer #1 (Recommendations for the authors):*

In this manuscript, the authors describe the earliest systematic differences in sex-specific splicing in *Drosophila* embryos or any animal for that matter. They find that differences arise already during the first few hours of embryogenesis and also identify a maternally-deposited pioneer transcription factor that contributes to generating these differences. The authors also provide a bioinformatics pipeline to analyze splicing over time.

The main strength of this paper is that the authors were able to generate pure populations of male or female embryos (using a recently published genetic system in *Drosophila*), and they exploited this by generating numerous genome-wide datasets. Their analyses revealed an interesting link between a maternally deposited transcription factor and alternative splicing, in particular, of genes that are differentially spliced between males and females.

A weakness of this paper is that the argument that CLAMP's effect on splicing is functionally meaningful is not fully substantiated by the data.

Whereas the observations that loss of CLAMP affects the splicing of a set of genes, many of which seem to be involved in sex determination, a number of other observations do not fit with the "master regulator of splicing" role for CLAMP that the authors are pushing.

For example, the authors show that in early embryos, where they detect xxx genes show CLAMP dependent splicing events, only 8-20% of these genes are actually bound by CLAMP. In later embryos they say 60-65% of genes affected by CLAMP are also bound, but CLAMP may be binding to a very large number of expressed zygotic genes at this time. There are no statistics to show that this overlap is meaningful. It is true that the pattern of CLAMP binding is different in different subsets of genes, but there is no concrete information of how many genes were used to generate the plots in Figure 3, making it difficult to evaluate the meaning of these data.

Moreover, the authors compare CLAMP binding to RNA and CLAMP-dependence on splicing for the two cell lines they use (as they only have CLAMP RNA binding data for the cell lines). In these data, 452 genes show CLAMP-dependent changes in splicing, but only 54 are bound by CLAMP and the authors say only 10 genes are direct targets of CLAMP-mediated splicing regulation. The authors conclude that the rest of splicing regulation is due to mis-splicing of other splicing regulators and are thus indirect effects of CLAMP. If such a small fraction of binding sites correlate with splicing changes, how can we interpret other analyses that take into account all CLAMP binding sites?

Similarly, the overlap between CLAMP and MLE binding is minimal, yet the authors conclude that CLAMP "sequesters" MLE and prevents it from binding at specific sequences. But there seems to be a lot of MLE binding that is completely independent of CLAMP in wt, so it's unclear how the authors propose CLAMP is preventing MLE from undesired binding.

An original concern was that even though the relatively minor effects of CLAMP on alternative splicing are an interesting observation, there is no indication of the functional significance of these changes. The authors claimed to have addressed this, but this is not the case. There is still no indication that any of the changes in splicing are functionally significant. I don't mean to say that it is not important to document these changes, but all claims of functionality are not supported by any piece of data. The key regulator of sex determination, sxl, whose splicing is somewhat changed, is affected by the loss of CLAMP in a much stronger way at the protein level than at the alternative splicing level. It is thus not fair to say "We demonstrate the functional significance of CLAMP-dependent alternative splicing by determining that CLAMP-dependent changes in sxl splicing in females induce the formation of the male-specific lethal dosage compensation complex in females that never normally occurs". This could all be due to the extremely reduced level of sxl at the protein level.

Another concern was the question of uncoupling the effects of CLAMP on transcription and splicing. The authors provide some comment that 85% of genes affected at the level of splicing are not regulated at the level of transcription, but there is no data shown or any details as to how this comparison was done.

Overall, the paper has not changed much from the previous version we reviewed. The authors present a very large amount of data. These are not always totally clear and often seem over-interpreted. I still do not think that the strong push for a role of CLAMP as a master regulator of splicing is substantiated.

*Reviewer #2 (Recommendations for the authors):*

The authors of this manuscript have added significant additional data to support a role for maternal Clamp in the regulation of sex-specific alternative splicing. These data, while correlative, help to convince me that regulation of alternative splicing is a major function of Clamp. Intriguingly, the new data present yet another mechanism of Clamp-dependent gene regulation through 5'UTR association and FMRP-dependent regulation of translation. This surprising finding helps put to rest a concern from the previous version of the manuscript, where the reduction of Sxl protein levels seemed to poorly correlate with the magnitude of the change in splicing observed. As such, it would seem Clamp's major role in Sxl regulation occurs at the level of translation control, as opposed to splicing or transcription initiation. An additional complication that will warrant future investigation.

The manuscript is dense and completely packed full of data and analyses. This strength is also its flaw, as it remains challenging to follow the thread of the story at times as the models and assay systems change. Nevertheless, I feel that it is important that the work should be published without further delay so that others may benefit from the discoveries and data sets described in this work.

I have a few suggestions for the authors about ways to help clarify the presentation.

1. It would be wonderful if the figure legends would include the identity of the assay used to collect the data. For example, the legend for Figure 4E does a great job of this, but Figure 3, the rest of Figure 4, and Figure 5 would benefit from the same level of detail. This saves the reader from jumping back and forth so much between the text, the figure, and the legend while trying to understand the data.

2. The rationale for using specific statistical tests should be presented in the manuscript and/or legend. Perhaps a section in the methods for statistical analysis? For example, Figure 1D relies upon a chi-square test (why not ANOVA?) while Figure 2A relies upon Fisher's exact test. I think I understand why (sample size), but I'm guessing. Figure 2E the p-value range is not clear for the left and center panels.

3. Did the authors mean to use "transcriptions" on the left axis of Figure 2A graph or "transcripts"?

4. The 5'UTR in figure 7C should be labeled.

*Reviewer #3 (Recommendations for the authors):*

In this manuscript, the authors carry out a detailed examination of sex-specific gene expression and pre-mRNA splicing during early *Drosophila* embryogenesis. They take good advantage of a meiotic drive system that had been previously implemented in the PI's lab to enable the collection of sufficient amounts of properly sexed embryos to perform various genomic assays. The authors carry out transcriptome (RNA-seq) and chromatin (Cut&Run) profiling experiments in the presence or absence of a maternally provided transcription factor, called CLAMP (chromatin-linked adaptor for MSL proteins). They analyze two different developmental time points (0-2 hrs and 2-4hrs after egg laying) corresponding to pre- and post-ZGA (zygotic gene activation) embryos, respectively.

This is a huge manuscript (80+ pages) with a ton of supplemental figures and data. There is a lot to like here, but in my view, the manuscript needs further revision. Most of my critiques can be addressed without further experimentation. There is a good story here but I feel that the stronger points get diluted by the weaker arguments.

Response to Previous Review

I did not participate in the previous round of review and so have tried to avoid bringing up new points that were not raised in the first round. Rather than diving into the details straightaway, I would say that the main criticism raised by the referees was one of data over-interpretation. Personally, I am not comfortable making deep mechanistic conclusions (e.g. an association with the catalytic step 2 spliceosome) largely on the basis of genome-level analyses. After reading the revised manuscript and the response to the review I still feel that some of the data are being pushed beyond their limits. The authors' model may well be correct, but the narrative in many parts of the manuscript goes from a given finding being what I would say is "consistent with" a certain interpretation, rather than one that actually "suggests" it works that way.

General points

1. CLAMP is a general transcription factor, but it has a well-documented role in the histone locus body (HLB), located at the histone gene complex (HisC). Reduced expression of histones can have major effects on gene expression (on both transcription initiation and downstream RNA processing steps). The potential for pleiotropic effects on transcription (e.g. elongation rates are known to affect splicing) due to reduced histone dosage is not really mentioned.

2. Line 142. Claims of primacy should be removed from the Results section. That sort of thing can be used in an introduction summary or in the discussion of the results. Using it as a conclusion in the Results section ("Therefore, we defined sex-specific splicing events in the early embryo for the first time.") just seems a bit odd.

3. Some aspects of the Results need to be reworked. Probably got mixed up in the revision. As it now stands, the subsection starting on line 150 is redundant and out of sequence. There is a whole list of reasons for doing these experiments in the Intro (lines 78-87), seems like line 150 starts to make the same arguments over again. Furthermore, the information on lines 154-159 really should have been introduced on/near line 128 where the authors first present results of splicing analysis following CLAMP depletion.

4. Line 188. The way that this sentence is written, the authors have already concluded that CLAMP regulates splicing. At this point in the narrative, loss of maternal CLAMP could affect SSS by any number of indirect means. "Regulation" implies something more active. So I'm not sure you can say start off with: "Furthermore, 85% of genes at which clamp regulates SSS…" because it assumes facts that are not in evidence. I apologize if this comment sounds picayune but this sort of logic matters when you are building an argument.

5. The Cut&Run data in Figure 6 are curious. I worry that there is some sort of normalization problem with the dataset. In the peaks that were identified in the male control embryos (panel A), roughly two-thirds of the sites on the autosomes and half the sites on the X chromosome are essentially flat. Does that mean the peak that was called by MACS2 is really more than 1.5 kb wide? Maybe that makes some sort of sense on the X. But on the autosomal sites, it looks like noise. The signal in the flanking regions next to the sharper peaks in the second heatmap column (Male control) looks too high. Well above the background binding levels in all of the other columns. This suggests that maybe there are simply more reads in this sample. I cannot tell without seriously digging into it. Is MLE really coating large chunks of chromatin on the autosomes?

Why is there such an abrupt transition from the set of narrowly defined peaks to a set of wide, shallow ones? If there were some sort of second criterion one could use for peak calls then you might be able to exclude (or include) certain regions. Right now it just looks like autosomal noise.

Again, for the autosomes at least, it might make some sense to try and find a common set of peaks that are found in both the male and female control samples. Then look at the effects of CLAMP depletion on that subset. In addition to a heatmap, one could use DESeq2 to quantify the difference in a metaplot for males vs females (+/- clamp).

Specific ideas for revisions:

Figure 1. Panels A and D. The multiple uses of various shades of gray in panel D versus similar shades of gray in panel A are confusing. This could be improved with color to make it a bit more readable. In panel A, I suggest that the authors use a unique color for each of the "differential" exons (currently they are all in black) examined in the 7 classes. Then shade the corresponding bars in panel D with that same color (one of which could be black). That way you can continue to use gray in panel A for the exons that are not differentially analyzed.

Figure 2E and lines 212-214. Most of the SSS genes in the early female embryo encode transcription and splicing factors? This statement is not at all obvious or even well supported by Figure 2E. There are two dots, one roughly 3 genes and the other 6-7 genes? What is the numerator? What is the denominator? Why should I believe this finding is significant? Why is the adjusted p-value bar all one shade in the two on the left? More stat power in the third GO term panel? I feel like the major points being made in Figure 2A-B get diluted with the additional panels and Venn diagrams. Better to focus the reader on solid conclusions.

Lines 239-274. Three full paragraphs of text are assigned to Supplementary figures. Does that not seem excessive? If the RNA-seq data regarding zygotic CLAMP expression are not going to be presented in the main body figures, why so much text? This leads to Figure 3.

Figure 3. Important points are being made here, but I fear some of the points are getting lost in the blizzard of metaplots. I don't have any good suggestions for how to streamline but it seems if a few key points could be distilled out of Figures3, and from the supplementary tables cited in the three paragraphs above (lines 239-274), that a single main-body figure with most important points would be impactful.

Figure 4. This whole figure should be reworked and most of it sent to the supplement. Panel E should be deleted. The information content in panels C and D is really low. That leaves A and B. What are the points being made in these panels? I don't think that the authors make much out of the motifs in the text. So the main points.

Figure 5 (see general points above). Reanalysis seems to be in order.

Figure 6. This figure may need some reshuffling or even split in two. I think that most of the readers of the paper will be confused by the fact that the males do not show any male-specific splicing in panel B. After reading the nuanced text (lines 519-520) a couple of times, where the authors mention that the embryos have not yet become "fully specified," I realized that this is actually the expected result. Maybe the authors should lead with that. Or be more explicit. In fact, I'm not even sure the pre-ZGA transcripts of Sxl are even translated in the early embryo. But that's a story for another time.

In this part of the manuscript, I don't think the reader is quite ready for panel A, which could be combined with panels D and E to make a new figure. Meanwhile, it might be helpful to bring back two of the panels from the older version of this figure (currently in Figure S11). The gene model in the current panel C (X-axis) is poorly annotated and the Y-axis is unlabelled. I found the panel particularly unhelpful and had to look at Figure S11 to figure out what was going on. I suggest bringing back panels S11A and S11F into the main body somehow. This would show the casual reader that later on in development splicing works the way it is depicted in all the textbooks. Then explain to the reader that early embryos have, by definition, maternally spliced Sxl transcripts. The authors have RNA-seq data that for these time points, why not use them? Analysis of splice-junction reads in the RNA-seq could be added to flesh out an entire figure about Sxl splicing.

A new figure could be used to show the pathway and the splicing of Tra, *Dsx*, Msl2, etc.

Figure 7. The overall model. I don't know what to say about the current Tfigure other than it's pretty complicated. The biology is complex, so I get it. But am worried that the authors' main points are going to be lost on a readership that will not appreciate all the nuances.

---

## [Author Response]

[Editors’ note: the authors resubmitted a revised version of the paper for consideration. What follows is the authors’ response to the first round of review.]

Comments to the Authors:We are sorry to say that, after consultation with the reviewers, we have decided that this work will not be considered further for publication by eLife.The three reviewers agreed on a number of points, with the main ones being that:1. The mechanistic conclusions on how CLAMP may affect alternative splicing are not fully substantiated by the data;2. That the effect of CLAMP loss of function on Sxl is quite minimal – despite the effect at the protein level appearing to be very substantial; this apparent discrepancy raises several concerns about what direct vs. indirect effects of manipulating CLAMP.The reviewers had several specific points in addition to this, which are attached below.Reviewer #1 (Recommendations for the authors):In this manuscript, the authors describe the earliest systematic differences in sex-specific splicing in *Drosophila* embryos or any animal for that matter. They find that differences arise already during the first few hours of embryogenesis and also identify a maternally-deposited pioneer transcription factor that contributes to generating these differences. The authors also provide a bioinformatics pipeline to analyze splicing over time.The main strength of this paper is that the authors were able to generate pure populations of male or female embryos (using a recently published genetic system in *Drosophila*), and they exploited this by generating numerous genome-wide datasets. Their analyses revealed an interesting link between a maternally deposited transcription factor and alternative splicing, in particular, of genes that are differentially spliced between males and females.A weakness of this paper is that several mechanistic conclusions are drawn from rather correlative experiments. So while the main observations are very interesting, the mechanistic model should be interpreted more cautiously.

We agree with the reviewer that it is always important to be cautious in the interpretation of genomic data, and therefore we have edited the discussion to highlight that we have identified multiple subsets of CLAMP-dependent alternative splicing events:

Events that are directly regulated by CLAMP on chromatin independent of Sxl that involve direct interaction between CLAMP and DNA and RNA of the target genes, including validated targets (Tables S6 & S10 and Figure S12A-D).Events that are directly regulated by both CLAMP and Sxl together (Table S10).Events that are indirectly regulated by CLAMP through modulating the function of Sxl and other RNA binding proteins, including CLAMP-interacting hnRNPs for which we have added new validation (Figure S13).

Events that are indirectly regulated by CLAMP likely arise because CLAMP regulates the alternative splicing of genes encoding other splicing regulators, including validated CLAMP targets (Table S2 and FigS4), thereby amplifying the direct effect of CLAMP. Although the number of targets at which CLAMP binds to both the DNA and RNA is not extensive, the direct CLAMP targets are often genes that are critical for regulating alternative splicing (Table S2 and S6-7; Figure 6, S4, and S11), suggesting that CLAMP functions as a master regulator upstream of key regulators of alternative splicing.

Therefore, we now discuss an additional model in which CLAMP regulates a subset of its targets by functioning upstream of Sxl as a master regulator of sex-specific splicing, consistent with its binding to chromatin, including the *sxl* locus very early in development (Duan et al., 2021).

We also now further highlight that evolutionary analysis suggests that the ancestral function of CLAMP is as a splicing factor, and its function in dosage compensation evolved from its splicing function (Quinn et al. 2016, Howard Chang’s lab).

In the discussion, we have also highlighted recent literature that has identified direct contacts between other TFs and RNA (Henninger et al. 2021, Sharp et al. 2022, Oksuz et al. 2022).

During the revision process, we have added several new data sets further supporting our model, including:

Alternative splicing analysis from cell lines (Table S5 and Figure S5) to match iCLIP data in cell lines and splicing data from L3 larvae (Table S3) to highlight the role of CLAMP in splicing during later developmental stages to support our larval validation data (Figure S11).Analysis of sxl splicing in embryos (Figure 6B).Motif analysis of RNAs that bind to CLAMP (iCLIP) suggesting that Sxl and hnRNPs are both cofactors for CLAMP (Figure S12A-D and S13).Validation of three additional CLAMP-dependent sex-specifically spliced isoforms that all encode essential RNA binding proteins, including functionally characterized and disease-linked transcripts (Table S2 and Figure S4).

This manuscript presents a lot of data exploring the question of regulation of sex-specific splicing by a maternally deposited factor, in early *Drosophila* embryos. These data show:– Loss of maternal CLAMP affects alternative splicing of 200-400 transcripts, with a trend to affect a bit more the mutually exclusive exon category.– This is a small fraction of the total alternative splicing events in embryos at these times, but a larger fraction of the sex-specific alternative splicing (30-60%).– Loss of maternal CLAMP causes both loss and gain of alternative splicing events.– ChIP-seq of CLAMP shows two types of distribution over genes: on genes whose level is affected by CLAMP, it is enriched at the ends of the genes; on genes whose alternative splicing is regulated by CLAMP, there is an enrichment over the whole gene body.– iCLIP of CLAMP shows binding to RNA, predominantly on chromatin (experiment done in cell lines). It binds mostly mRNAs but also other RNAs– CUT&RUN of the helicase MLE also has a dual type of enrichment over genes. Upon loss of maternal CLAMP, the signal in males looks globally lower, in females less so. But other peaks are also gained, suggesting a re-distribution.– CLAMP is necessary for sex-specific splicing of Sxl in a manner that correlates with chromatin accessibility over the alternative exon.

Thank you for these important points. We agree that not every association of CLAMP with RNA will be meaningful and have now added a further discussion of potential indirect functions in the discussion.

We also highlight the following points that support a direct role in sex-specific splicing:

CLAMP binds to snRNA sex-specifically on chromatin in males and not females, which makes it unlikely that proximity to chromatin alone drives this binding (Figure 4C).Additionally, CLAMP binds to protein components of the spliceosome sex-specifically (Figure S10), and its evolutionarily conserved binding sites are located at splice junctions across diverse species (Quinn et al. 2016).Furthermore, we have evidence supporting a direct role for CLAMP in splicing because a subset of genes at which CLAMP is functionally required to regulate sex-specific alternative splicing are also bound by CLAMP at both the DNA and RNA level in a sex-specific manner (Table S6 & S7 and Figure 6C and S11C, E).Also, we present a global view of the redistribution of the CLAMP-interactor MLE in males in Figure 5, demonstrating its extensive redistribution.

The observations presented here are very interesting, but a few critical aspects of the mechanism are over-interpreted. For example, I find it hard to know whether the association with snoRNAs found in iCLIP is meaningful. By the same criteria, we should conclude that CLAMP associates with tRNAs. I think it's more likely that the proximity of spliceosomes to chromatin produces the observed signal. Also, the specific focus on particular peaks of CLAMP and MLE is difficult to interpret without more global views of how these very general proteins redistribute along the genome. Also, the connection between MLE redistribution and alternative splicing is not really explored adequately. I don't think the authors need more data for this paper, but I would suggest more concise and accurate descriptions of the data and more cautious interpretations.Regarding the last experiments looking at splicing of Sxl, it seems unclear to me why the authors switched from the RNAi tool to ablate the maternal contribution, to a mutant allele which (if I understand correctly) ablates the zygotic contribution. It is also unclear why even though a small fraction of Sxl is mis-spliced in females, the effect on the protein level is so dramatic (!). This suggests that something else is happening in these embryos that lack CLAMP. This actually raises some doubt about what causes all the effects in the earlier part of the paper, but we cannot really compare these experiments given that the authors used very different genetic tools.

We agree with the reviewer that we need to more clearly justify the transition between discussing maternal and zygotic CLAMP and the tools used for ablation. Therefore, we have addressed this concern in the text by separating maternal and zygotic functions into different figures.

We also clarified the following methods that we used to disrupt maternal and zygotic CLAMP: (1) a maternal triple driver driving *clamp* RNAi was used to ablate the maternal contribution due to the location of the *clamp* gene being very close to the centromere, making germline clones extremely difficult. (2) The *clamp^2^* mutant is the most complete way to ablate the zygotic contribution, and therefore, this was chosen to examine the function of zygotic CLAMP.

In the revised manuscript, we have added new data to a sure that we are comparing within the same developmental stage or cell line whenever the techniques are feasible. For example, we have added RT-PCR analysis of *sxl* splicing showing how maternal CLAMP regulates the splicing of *sxl* transcripts in 0-2 and 2-4 Hr embryos (Figure 6B) which further validates our genomic analysis of sex-specific splicing at these same time points. Consistent with the ChIP-seq binding patterns for CLAMP at the gene loci at these time points (Figure 3), there is a stronger function for CLAMP in splicing at 2-4 hours than at 0-2 hours, likely due to the initiation of zygotic transcription at approximately 2 hours. In contrast to the much later larval stage, in embryos, the male and female isoforms of *sxl* have not become fully specified, consistent with the literature on autoregulation of Sxl (Horabin and Scheld 1996, Moschall et al. 2019).

Furthermore, we would have liked to perform iCLIP in sexed embryos, but the very large amount of material required for this approach makes it very challenging to perform with our meiotic drive sexing system, which produces a low yield of sexed embryos. Therefore, we included splicing data from cell lines that match our cell line iCLIP data (Table S5).

To increase clarity, we have separated the results showing the function of maternal CLAMP (Figure 6) from that of zygotic CLAMP (Figure S11) on the splicing of the components of the sex determination pathway. We have also highlighted in the text prior work from collaborators, which shows that CLAMP is required for regulating Sxl protein levels in early embryos which we show in larvae in this manuscript (Collonetta et al., 2021).

In addition, we agree with the reviewer that there is a difference in the magnitude of the effect of CLAMP on *sxl* splicing from that on protein levels. Therefore, we have now added additional data and interpretation that provides a possible explanation as follows:

iCLIP results show that CLAMP binds to the sxl transcript in females (Column F, Table S7) at the 5’ UTR region (Figure S11C). Therefore, we hypothesize that CLAMP binding might have a role in regulating the translation of the sxl transcript into protein, and thereby the loss of CLAMP could regulate both the splicing and translation of the sxl transcript.

It is possible that in the absence of CLAMP which binds to the 5’UTR of the sxl transcript, other proteins involved in translational repression may bind, preventing Sxl protein from being made. Most CLAMP is in the nucleus as expected, but the small amount of cytoplasmic CLAMP also associates with RNAs and RBPs. For example, CLAMP normally binds to the translational repressor FMRP (Fragile X protein) in the male but not the female cytoplasm, which we have now added to the manuscript (Figure S11D). Direct and indirect functions for CLAMP in both splicing and translation may explain how an effect on sxl splicing of a smaller magnitude is linked to a larger reduction in Sxl protein levels.

Reviewer #2 (Recommendations for the authors):Ray, Conard et al. describe the role of the CLAMP transcription factor in the regulation of sex-specific alternative splicing in *Drosophila* embryos, larvae, and tissue culture cells. The results are presented from three main lines of investigation: (1) genomics in early embryos, including sex-specific RNA-seq, which is analyzed with "time2splice" a newly developed pipeline for detection of alternative splicing events; (2) iClip in cultured S2 and Kc cells. (3) Validation of CLAMP's effect on alternative splicing of core sex determination factors Sex Lethal, Transformer, and Doublesex. There is very much data included in this manuscript, and the presentation of the genomics data will require substantial clarification for readers accurately to interpret the results. At the heart of the manuscript is the observation that clamp mutants demonstrate aberrant expression or loss of expression of sex-specific genes, and the argument presented here is that this is largely due to the misregulation of splicing in clamp mutants. While the genomics data suggest that CLAMP is necessary for certain sex-specific alternative splicing events and that CLAMP interacts with splicing factors, CLAMP appears to only have a small effect on the specific examples of sex-dependent alternative splicing (Sxl, Tra, Dsx) presented as validation. The findings also catalog for the first time the splice variants present at the maternal-to-zygotic transition, but the current analysis of these data leaves open the question of whether such alternative splicing events are associated with zygotic transcripts, and whether the magnitude of CLAMP's effect in this process is significant.

We thank the reviewer for their important comments, and we have addressed them in more detail below.

To summarize:

We demonstrate the functional significance of CLAMP-dependent alternative splicing by determining that CLAMP-dependent changes in sxl splicing in females induce the formation of the male-specific lethal dosage compensation complex in females that never normally occurs.We validated additional alternative splicing events beyond those at the sxl, tra, and dsx, genes, including seven direct CLAMP target isoforms bound at both the DNA and RNA level on chromatin that has established functions in splicing and development (Table S2 and Figures6, S4, S11). For example, Fus is a regulator of alternative splicing, and CLAMP-dependent sex-specific isoforms of fus have known functions (Table S2).We emphasize that evolutionary evidence suggests that the ancient function of CLAMP is as a splicing factor (Quinn and Chang, 2016).

– The title of the manuscript touts the regulation of alternative splicing by a maternal pioneer factor, but a substantial proportion of the data is derived from cultured cells or third instar larvae, where the maternal contribution of CLAMP is not substantial. It is also unclear what pioneering per se has to do with the mechanism the authors propose and what little connection they present is not focused on in any validation or mechanistic follow-up work. I recommend changing the title to de-emphasize CLAMP's maternal expression and pioneer activity.

Thank you for the comment regarding CLAMP as a pioneer factor in the title of our manuscript. We agree that we have already shown that CLAMP functions as a pioneer factor (Duan et al., 2021), but the key link that we want to emphasize here is that we have identified a new function for a pioneer factor in splicing has not yet been studied for other pioneer factors.

We have shown that CLAMP regulates chromatin accessibility at the key *sxl* exon 3, whose splicing is regulated by CLAMP, providing a link between the CLAMP pioneering activity and splicing. Furthermore, we show that at the stages when CLAMP functions as a pioneer factor (04 Hr embryos), CLAMP binds to sex-specifically spliced genes along gene bodies near intronexon junctions (Figure 3A-D and Figure S6), which is different from its more TSS/TES specific localization on chromatin at genes that it regulates transcriptionally (Figure 3E-H, Reider et al. 2021).

The binding of a pioneer TF near intron-exon boundaries, its function in alternative splicing, and direct interaction with target RNAs, spliceosomal RNAs, and RBPs provide a new mechanism that explains observations in that literature suggesting a link between chromatin accessibility and splicing mediated by an unknown mechanism (Agirre et al. 2021, Petrova et al. 2021).

To address the important concern that our study does not sufficiently address the embryonic function of CLAMP, we now emphasize that most of our study focuses on how maternal factors regulate RNA processing as the embryo develops (Figures1-3 and 5-6). To improve the clarity of our presentation, we have now separated the zygotic function of CLAMP into a supplementary figure and tables (Figure S11 and Table S3, S5). In this way, we emphasize that most of our work focuses on how depleting the maternal reservoir of CLAMP (using the *MTDGAL4* driver) regulates sex-specific splicing during the earliest developmental stages (0-4 hr embryos). We have also added additional analysis of *sxl* splicing in embryos (Figure 6B) as described above.

The experiments that were performed outside of embryos were done in these contexts for technical reasons because we were unable to collect enough sexed embryos from our low-yield meiotic drive system to perform them, such as iCLIP. We have added sex-specific alternative splicing data from cell lines to match the cell line iCLIP data we generated (Figure S5 and Table S5).

– Throughout the manuscript, but particularly within the description of the RNA-seq results, the number of objects (genes, transcripts, splicing events, peaks, etc…) are rarely stated explicitly in the text, but are instead sometimes part of figures or legends, if they are stated at all. Please edit the manuscript throughout to indicate the number of objects being compared and include percentages of the relevant group when discussing specific categories (e.g., x% of total genes/transcripts are alternatively spliced (n/N). Y% (n) of alternatively spliced genes/transcripts are sex-specific. Of these z% (n) are zygotically expressed.). As written, I was unable to clearly evaluate the biological conclusions for the first half of the manuscript because I had to rely on non-quantitative descriptors provided by the authors to glean magnitudes: for example, "very low levels," (line 260).

We agree with the reviewer that it is important to add exact numbers of objects in the figures and text and not just in methods and supplementary tables where we included them previously.

Therefore, in the revised manuscript, we have added the values for the numerator and denominator in each category in the text and in the revised figures (Figure 1 and 2, Figure S2). We have also added a chart and bar plot (Figure 1C-D) to show the number of alternative exons affected in each category. The sex-specific splicing events/transcripts were identified by comparing deltaPSI values between different categories classified by sex and the presence or absence of CLAMP (Figure S2, Figure 2A-B). The genes to which these transcripts belong are listed in Table S6. The number of CLAMP-dependent spliced genes in each category is shown in Figure 2D.

– In terms of the magnitude of the effect: piecing together information in Figures 1 and 2, there appears to be 10891 total (genes? transcripts?) in the 0-2 hour female RNA-seq data. Figure 1B implies that 16.27% (1771) of these are alternatively spliced. Can the authors comment on what an expected range of alternative splicing would be for a 'typical somatic cell' of any sort?

We have addressed this question below by adding a new table within Figure 1 to highlight the number of alternatively spliced exons we have identified in control samples (Figure 1C). This number is comparable to what has been previously identified from other *Drosophila* species (Gibilisco et al., 2022).

– For Figure 2D and associated text, the authors address whether CLAMP-dependent sex-specific alternative splicing is observed mainly in zygotic genes. I have several issues here. Mainly, I am unclear on how this comparison is being made (partly because of the lack of numbers in the text). The numbers of sex-specific AS genes in the legend are different than the numbers in the Venn diagrams. From the minimal explanation of how this was done, the impression is given that if an AS gene was not one of the 841 maternal genes, then it is likely to be zygotic. This does not follow logically, given that most of the 0-2 hour transcripts (~10891?) will by definition be maternal, and that non-membership in this limited list of 841 maternal genes (from which source?) cannot directly imply that the gene is zygotic, since most of the remaining transcripts will be maternal but not included in the limited list of 841. There are near-exhaustive gene lists of purely zygotic genes (DeRenzis/Wieschaus), and maternal-zygotic genes (Chen/Zeitlinger, Kwasnieski/Bartel). This analysis may be enhanced by enumerating the fraction of purely zygotic genes and maternal-zygotic genes. This is an important analysis and should be (1) re-done, and (2) documented extensively (with accurate numbers).

We agree with the reviewer that it is important to carefully compare our splicing analysis with previous lists of maternal and zygotic transcripts generated by multiple laboratories. We had part of this analysis in the original manuscript but have now improved this analysis as the reviewer suggested.

To clarify our analysis, 10,891 is the total number of exons alternatively spliced in 0-2 Hr female embryos out of a total of 66,927 exons (new Figure 1C). The remaining exons, (56,036 exons) are constitutive exons that are present in all transcripts of the genes they belong to and are not alternatively spliced.

Δ PSI values quantify individual splice junctions by measuring the difference in the percent spliced in (PSI) for a particular exon using PSI=IR (included reads)/ IR+ER (excluded reads). The difference in PSI values (DeltaPSI) between samples quantifies the differential inclusion or exclusion of alternative exons between the two sample types (Methods). Using this quantification method, we determined the significant CLAMP-dependent splicing events (FigS2C-F). Also, the CLAMP-dependent sex-specific splicing events (transcripts) are listed in Table S1 (Figure 2A-B), and the genes they belong to are listed in Supplementary Table S6.

We have also improved our comparison with lists of maternal genes as follows:

We compared the maternal genes from Kwasnieski/Bartel 2019, which integrates lists from all prior publications at three different developmental stages: (1) NC9-10, (2) Syncytial blastoderm, and (3) Cellular Blastoderm (Figure 2D) with our female and male sex-specifically spliced gene lists at the pre-MZT and post-MZT stages (Table S6). As expected over time, the three developmental stages show a gradual decline in maternal transcripts. Therefore, we compared our pre-MZT splicing list with the NC9-10 and syncytial blastoderm stages (Figure 2D, first and second Venn diagrams) and the post-MZT list with the syncytial and cellular blastoderm stages (Figure 2D, third and fourth Venn diagram). We found very low levels of overlap of CLAMP-dependent spliced genes with genes having maternally deposited transcripts (Figure 2D). Only 19.3% (23/119) of female and 22.5% (22/98) of male CLAMP-dependent 0-2 Hr pre-MZT genes overlap with maternal genes with transcripts at NC-19, which drops to 12.6% (15/119) and 12.2% (12/98) respectively when compared to syncytial stage maternal gene transcripts. Similarly, only 22.2% (46/207) of female and 24.5% (26/106) of male CLAMP-dependent 2-4 Hr post-MZT genes overlap with maternal genes with transcripts at the syncytial blastoderm stage, which drops to no overlap when compared to maternal transcripts at the cellular blastoderm stage.

– Corollary to the above point: the 10891 number (Figure 2A): does that refer to unique transcripts (tx) or unique genes (gn)? The enumeration requested in the above point should clearly state that the numbers are tx or gn, and if tx, the total number of gn represented by that value should be cited. Comparisons with published gene lists should be done in the appropriate 'unit' (tx or gn), dictated by the published list.

We agree with the reviewer that it is important to clarify whether we are quantifying genes or transcripts. 10,891 (Figure 2A) is the total number of exons that undergo alternative splicing. Our time2splice pipeline uses an exon-centric approach to identify affected splicing events and is based on a standard method in the splicing field called SUPPA (Methods). We have added a chart and bar plot (Figure 1C-D) to show the number of alternative exons affected in each class. The sexspecific splicing events/transcripts are identified by comparing deltaPSI values between different categories depending on sex and the presence or absence of CLAMP (Figure S2, Figure 2A-B). The genes to which these transcripts belong are listed in Table S6. We have added the number of CLAMP-dependent alternatively spliced genes in each category directly into Figure 2D.

**Author response image 1. sa2fig1:** Maternal CLAMP regulates sex-specific alternative splicing during early embryonic development. Gene Ontology (GO) results for genes showing CLAMP-dependent female sex-specific splicing in embryos at the 0-2 Hr pre-MZT stage and for genes exhibiting CLAMP-dependent female and male sex-specific splicing in embryos at the 2-4 Hr post-MZT stage. The size of the circle increases as the number of genes in that category increases. The color of the circle represents significance (p-value). GO categories for male embryos at the 0-2 Hr pre-MZT stage are not shown because the gene set is small, and no enriched GO categories were identified.

– The authors should comment on how a maternally supplied transcript could be alternatively spliced in a sex-specific manner. At least some of the sex-specific alternatively spliced genes are identified as being maternal in Figure 2. Are these maternal genes that are also zygotically expressed? Are the maternal isoforms consistent with the female zygotic isoform? Any detection of sex-dependent alternative splicing in solely maternal genes (i.e., not zygotically expressed) could indicate issues with the computational approach for scoring AS events and should be discussed.

We agree with the reviewer that it is important to clarify that the target genes that are regulated by CLAMP are zygotically transcribed, which is what our re-analysis of overlaps with maternal vs. zygotic genes demonstrates (Figure 2D). Furthermore, constitutively transcribed maternally deposited genes like CLAMP are often expressed in the zygote. The depletion of maternal CLAMP in this study was achieved using *MTD-GAL4>CLAMPRNAi,* which results in the depletion of CLAMP RNA in the mother's ovary, thus reducing protein levels in the ovary and therefore less CLAMP is deposited by the mother into the embryo. Therefore, it is possible that reduced levels of CLAMP in the ovary could affect the splicing of transcripts in the ovary, which are deposited in the egg, which we could detect. In this way, regulation of splicing of maternal transcripts by CLAMP could be functionally significant and does not have to be a technical artifact. In fact, we do observe regulation of maternal transcripts, although much less frequently than that of zygotic transcripts (Figure 2D).

– One of the differences between Kc and S2 cells is their sex, but this does not mean that any difference observed between the two cell types is sex-specific. The section beginning at line 335 reads as if 100% of the differences between Kc and S2 cells is interpreted as a sex-specific difference. In any case, I also felt that the results from this section should be confirmed in embryos somehow, perhaps through an RNA IP experiment. snRNAs should be abundant enough that they can be detected in a CLAMP IP, even from early male or female embryos.

We agree with the reviewer that it is important to highlight that Kc and S2 cells are different cell lines and have now highlighted this in the text. We would have preferred to generate iCLIP data from sexed embryos if it were feasible. Unfortunately, our sexed embryo system is low yield, and therefore, it was not possible to obtain enough material to perform IPs from embryos despite our attempts. Moreover, comparisons between Kc and S2 cells have been used to study sex differences in the context of dosage compensation for over thirty years in dozens of papers (reviewed in Gelbart et al., 2009), and the cell lines are both hemocyte-like (Cherbas et al. 1994, 2014).

– Section beginning line 514: In general, one of the weaknesses of this paper is that it switches between embryos, larvae, and cultured cells, without much critical evaluation of whether such different contexts impact the strength of the conclusions. In this case, I am puzzled why the authors choose to solely rely on MNase-seq in cultured cells to make a point about CLAMP-dependent chromatin accessibility when they have also performed ATAC-seq on CLAMP-knockdown embryos. The observation that loss of CLAMP leads to a greater amount of accessibility at Sxl exon 3 specifically in Kc cells (which is presumed to be because these are female cells but could instead be a cell-type specific effect independent of sex). Such a large effect should be evident in mixed-sex embryo collections from a CLAMP knockdown, and these data should be shown and presented in the Results. Are sex-specific differences in CLAMP binding observed by ChIP-seq at this locus? This data would be essential to show as well.

We agree with the reviewer that it would be ideal to perform all approaches in sexed early embryos using our meiotic drive system. However, this is very challenging, as described above, due to the small number of embryos that can be generated compared to the amount of material required for biochemical approaches like iCLIP. Therefore, we have generated new data and new analysis of sex-specific splicing from cell lines to match our iCLIP data as well as splicing analysis of new larval data. We have also separated the data on zygotic CLAMP into a supplemental figure (Figure S11) and rewritten the text to highlight when we are discussing maternal vs. zygotic CLAMP, which both function in sex-specific splicing.

Furthermore, we have added IGV screenshots showing sex differences in CLAMP binding at the *sxl* locus (Figure 6C and S11E). Also, as the reviewer suggested, it would be ideal to have ATACseq data from sexed embryos, but our ATAC-seq data is from mixed sex embryos (Duan et al. 2021, #GSE152596), making it challenging to identify sex-specific changes. This data is not ideal for studying such sex-based changes. Hence, we have relied on the sexed cell line MNase-seq data and not the mixed embryo ATAC-seq data. However, both data sets show that CLAMP regulates chromatin accessibility (Urban et al. 2017, Duan et al. 2021).

– The magnitude of the effect of CLAMP loss of function on Sxl splicing, however, does not seem to be very large, given the near absence of Sxl protein in female larvae. Can the authors clarify how they interpret this discrepancy? The same could be said for the MXL results. Is the regulation of splicing only a minor function of CLAMP?

We agree that the effect of CLAMP on *sxl* splicing is not the same as that on protein levels. We hypothesize that the binding of CLAMP to the 5’ UTR of the *sxl* transcript and the male-specific interaction of CLAMP with the translational repressor FMRP (Figure S11C, D) may explain this discrepancy which we have now highlighted in the text. We now add additional data and interpretation that provides a possible explanation as follows:

iCLIP results show that CLAMP binds to the *sxl* transcript in females (Column F, Table S7) in the 5’ UTR region (Figure S11C). Therefore, we hypothesize that CLAMP binding may regulate the translation of the *sxl* transcript into protein, thereby amplifying a smaller effect of CLAMP on splicing into a larger effect on protein levels.It is possible that in the absence of CLAMP binding to the 5’UTR of the *sxl* transcript in *clamp* RNAi females, other proteins involved in translational repression may bind, preventing Sxl protein from being made. Most CLAMP is in the nucleus as expected, but cytoplasmic CLAMP also associates with RNAs and RBPs. For example, CLAMP normally binds to the translational repressor FMRP (Fragile X protein) in the male but not the female cytoplasm, new data which we have added to the manuscript (Figure S11D). Direct and indirect functions for CLAMP in both splicing and translation may explain how an effect on *sxl* splicing of a smaller magnitude is linked to a larger reduction in Sxl protein levels.

Several lines of evidence support that the regulation of splicing is a key function for CLAMP:

CLAMP binding sites have been shown to have evolved from splice junctions (Quinn et al., 2016).Direct splicing targets of CLAMP that are bound at the DNA and RNA level on chromatin are often key regulators of splicing and development where specific isoforms often have different functions (Table S2, 3, 5-7 and Figures4, S7).A direct functional consequence of CLAMP regulating splicing is the induction of MSL complex binding to polytene in females that is never normally present (Figure S11I) due to CLAMP regulating the splicing of the *msl-2* transcript.The effect of CLAMP on MLE redistribution, a known component of the splicing complex, is quite large (Figure 5A, C, and D), and it can be visualized on polytene chromosomes (not shown).

– I have not yet reviewed the supplemental tables for completeness and suitability for use in follow-up studies. This should be revisited during the consultation.Reviewer #3 (Recommendations for the authors):In flies, it is well established that sex determination is controlled by gender-specific alternative splicing of the sxl gene. The current study extends the catalog of embryonic alternative splicing events, including numerous new gender-specific splicing events. The primary data set is RNA-seq from pre- and post-zygotic gene activation gender-specific embryo samples collected using the meiotic drive. The study identifies 92 transcripts differentially spliced between genders at 0-2 hours post fertilization, and 138 at 2-4 hours post-fertilization. A small subset (4) of splicing events were validated by alternative methods. In general, these data are convincing, though more validation would strengthen confidence in the data set.

To address this important concern, we validated four additional CLAMP-dependent sex-specific splicing events, which we added to Figure S4 and Table S2. We also validated the splicing of *sxl*, *dsx,* and *msl2* (Figure 6 and Figure S11).

The data are then compared to existing RNA-seq where maternal CLAMP is depleted. CLAMP is a candidate maternally deposited alternative splicing regulator. Between 30-50% of gender-specific splicing events are CLAMP-dependent, while only 2-3% of total splicing events are CLAMP-dependent. The authors suggest these results indicate a specific role for CLAMP in regulating gender-specific alternative splicing. The overall number of CLAMP- and gender-specific alternative splicing events is fairly low (<50) compared to the total number of alternative splicing events detected (>10,000). Having said that, the enrichment is significant by Fisher's Exact Test, and subsequent binding experiments provide additional support for the model.Next, the authors performed gender-specific ChIP-seq to map positions of CLAMP binding in embryos as a function of the developmental stage. Approximately half of the CLAMP-dependent gender-specific alternative splicing events show CLAMP enrichment on the DNA of alternatively spliced genes. This suggests (but does not prove) that CLAMP could be directly regulating co-transcriptional alternative splicing of gender-specific events. CLAMP also binds to some gender-specific mRNAs revealed by iCLIP data sets from male and female cell lines. This binding is enriched in RNAs that co-fractionate with chromatin, suggesting that DNA and RNA precipitation is coupled. CLAMP also appears to bind to snRNAs and components of the spliceosome, and loss of maternal CLAMP causes redistribution of the MLE complex, especially in male samples.Finally, the authors show that CLAMP plays a role in regulating sxl alternative splicing, and this role correlates to CLAMP-dependent chromatin formation.All told, the experiments point to a model where CLAMP regulates alternative splicing for a limited subset of embryonic transcripts, about half of which are gender-specific alternative splicing events, through a mechanism that involves DNA binding, RNA binding, and chromatin conformation. This is a fascinating outcome for a variety of reasons, as the concept of broad parental control over progeny splicing patterns has not been widely explored. It seems clear that the mechanism proposed accounts for a small fraction of the observed embryonic alternative splicing. There is no evidence that the novel alternative splicing events detected are important for embryogenesis or sex determination. Nevertheless, the results do move the field forward and provide testable hypotheses that can be addressed in future studies.In all cases where data is presented as a percentage, it would be MUCH CLEARER if both the numerator and denominator were presented, along with a p-value. For example, on line 279, it states "43.8% of all CLAMP-dependent sex-specifically spliced genes are bound by CLAMP:" This should be followed with (num/denom, p-value XXX) so it is clear to the reader that this percentage is meaningful and how large (or small) of a list of genes it describes. This should be done throughout the manuscript.

We thank the reviewer for this comment and have added numerators, denominators, and p-values for all percentages throughout the manuscript.

Several key experiments are relegated to supplementary information (for example, the volcano plots in figure S2). Where possible, critical experimental data should be moved into the body of the manuscript and confirmatory analysis placed in the supplement.

Figure 2B summarizes the key results from the volcano plots in Figure S2 and gives the exact number and distribution of CLAMP-dependent sex-specific events in different sexes and developmental stages compared to non-sex-specific CLAMP dependent splicing events. Therefore, we have placed the volcano plots in the supplement to streamline the main manuscript.

Some of the rationale was missing from the text, for example, the chromatin fractionation for the iCLIP data sets. I think I understand why this was done, but it should be presented.

We have added more rationale regarding the iCLIP chromatin fractionation which was done in order to identify target genes that are co-transcriptionally spliced and bound at both the DNA and RNA level. We have added the following lines to the manuscript (lines 342-345)

“We identified CLAMP RNA binding targets separately in chromatin and nucleoplasmic cellular fractions to understand whether CLAMP binds to both DNA and RNA at a subset of the same targets. Also, CLAMP RNA targets on chromatin are the most likely to be regulated by cotranscriptional RNA processing.”

The paper would be stronger with functional studies to assess the biological importance of the CLAMP-dependent sex-specifically spliced genes, although the manuscript is already overloaded with experiments and it strikes me as unreasonable to request more.

We agree with the reviewer that functional information is always important, and we do demonstrate an important functional consequence of CLAMP sex-specific splicing: preventing ectopic binding of MSL complex to the X-chromosome in females. We are pursuing the functions of additional targets which are often key regulators of alternative splicing (Table S2), but this is beyond the scope of the current work. We highlight the known functions of the targets and isoforms in Table S2.

The work as presented is difficult for the reader to get through. The paper would benefit from significant rewriting. I found significant overlap between the introduction and Results section, with several concepts and rationale presented in both sections. Also, in one case (concerning sxl) introductory material was first presented in the Results section. The reader would benefit from a more succinct presentation. In fact, the authors might wish to consider splitting the work into more digestible chunks, for example, a description of gender-specific alternative splicing events/stage-specific splicing events along with a more detailed description of the pipeline used to identify them, and a CLAMP paper with more functional characterization of the impact of CLAMP targets on embryogenesis. Ultimately, this decision is up to the authors, but I would ask them to consider it for readability/clarity's sake.

We have now removed two paragraphs of rationale from the introduction to eliminate redundancy. We appreciate the reviewer's suggestion to divide the paper into multiple stories. However, we decided to group our identification of sex-specific isoforms with the function of CLAMP to provide mechanistic insight into how sex-specific splicing is regulated beyond a descriptive paper about isoforms which we think is a stronger contribution to the field.

There are some typos that should be corrected (line 344 "most CLAMP RNA binds to hundreds of RNA", line 340 "Although CLAMP do not have a canonical RNA recognition motifs").

Thank you for finding these typos, which we have corrected.

[Editors’ note: what follows is the authors’ response to the second round of review.]

The manuscript has been improved but there are some remaining issues that need to be addressed, as outlined below:Reviewer #1 (Recommendations for the authors):In this manuscript, the authors describe the earliest systematic differences in sex-specific splicing in *Drosophila* embryos or any animal for that matter. They find that differences arise already during the first few hours of embryogenesis and also identify a maternally-deposited pioneer transcription factor that contributes to generating these differences. The authors also provide a bioinformatics pipeline to analyze splicing over time.The main strength of this paper is that the authors were able to generate pure populations of male or female embryos (using a recently published genetic system in *Drosophila*), and they exploited this by generating numerous genome-wide datasets. Their analyses revealed an interesting link between a maternally deposited transcription factor and alternative splicing, in particular, of genes that are differentially spliced between males and females.A weakness of this paper is that the argument that CLAMP's effect on splicing is functionally meaningful is not fully substantiated by the data.

To address this important reviewer concern, we have removed all claims that CLAMP directly regulates splicing. Instead, we have highlighted our identification of the first set of sex-specifically spliced transcripts in early embryos across species. However, we have included the effects of removing CLAMP on sex-specific splicing in embryos and larvae (Figures 5,6), we have changed our interpretation to describe the effects but do not conclude that they are direct.

To improve the clarity of our data presentation, we have:

Added the numbers of genes to average profiles, highlighting that CLAMP has a different binding pattern at genes where it influences sex-specific splicing vs. genes where it is involved in transcription (Figure 4).Moved a Venn diagram that highlights the lack of overlap between genes where CLAMP regulates splicing and those where it regulates transcription from the supplement to the main figure (Figure 3).

Also, we suggest in the discussion that defining the mechanisms by which CLAMP regulates splicing is an important future direction.

Whereas the observations that loss of CLAMP affects the splicing of a set of genes, many of which seem to be involved in sex determination, a number of other observations do not fit with the "master regulator of splicing" role for CLAMP that the authors are pushing.For example, the authors show that in early embryos, where they detect xxx genes show CLAMP dependent splicing events, only 8-20% of these genes are actually bound by CLAMP. In later embryos they say 60-65% of genes affected by CLAMP are also bound, but CLAMP may be binding to a very large number of expressed zygotic genes at this time. There are no statistics to show that this overlap is meaningful. It is true that the pattern of CLAMP binding is different in different subsets of genes, but there is no concrete information of how many genes were used to generate the plots in Figure 3, making it difficult to evaluate the meaning of these data.

We have removed the claim that CLAMP is a “master regulator of sex-specific splicing” in addition to the claim that CLAMP regulates splicing directly. We describe the data showing that altering CLAMP levels mis-regulates the splicing and potentially the translation of Sxl and several of its key targets, including *dsx* and *msl-2* (Figures 5, 6). However, we have removed the claim that CLAMP directly regulates Sxl splicing.

Moreover, the authors compare CLAMP binding to RNA and CLAMP-dependence on splicing for the two cell lines they use (as they only have CLAMP RNA binding data for the cell lines). In these data, 452 genes show CLAMP-dependent changes in splicing, but only 54 are bound by CLAMP and the authors say only 10 genes are direct targets of CLAMP-mediated splicing regulation. The authors conclude that the rest of splicing regulation is due to mis-splicing of other splicing regulators and are thus indirect effects of CLAMP. If such a small fraction of binding sites correlate with splicing changes, how can we interpret other analyses that take into account all CLAMP binding sites?

We agree with the reviewer and have removed all claims that CLAMP directly regulates splicing. Therefore, we have removed the iCLIP data from the manuscript and plan to perform additional experiments and further detailed analysis to explore how CLAMP interacts with RNA in vivo and in vitro in the future.

Similarly, the overlap between CLAMP and MLE binding is minimal, yet the authors conclude that CLAMP "sequesters" MLE and prevents it from binding at specific sequences. But there seems to be a lot of MLE binding that is completely independent of CLAMP in wt, so it's unclear how the authors propose CLAMP is preventing MLE from undesired binding.

We have removed this data from the present manuscript and plan to do additional experiments and further detailed analysis to explore CLAMP-MLE interactions in the future.

An original concern was that even though the relatively minor effects of CLAMP on alternative splicing are an interesting observation, there is no indication of the functional significance of these changes. The authors claimed to have addressed this, but this is not the case. There is still no indication that any of the changes in splicing are functionally significant. I don't mean to say that it is not important to document these changes, but all claims of functionality are not supported by any piece of data. The key regulator of sex determination, sxl, whose splicing is somewhat changed, is affected by the loss of CLAMP in a much stronger way at the protein level than at the alternative splicing level. It is thus not fair to say "We demonstrate the functional significance of CLAMP-dependent alternative splicing by determining that CLAMP-dependent changes in sxl splicing in females induce the formation of the male-specific lethal dosage compensation complex in females that never normally occurs". This could all be due to the extremely reduced level of sxl at the protein level.

Based on the reviewer's suggestions, we have rewritten the manuscript to remove the claim that CLAMP directly regulates splicing (Lines 412-418).

“In the absence of CLAMP, ectopic MSL2 protein (in red) is present at several locations on female chromatin in contrast to control females (*clamp^2^/CyO-GFP* heterozygous females) where the malespecific MSL-2 protein is not present on female chromatin as expected (Figure 6G). Similar to *dsx*, the msl-2 gene is also bound by CLAMP (Figure 6H) and regulated by Sxl and, therefore, could be regulated through both direct and indirect mechanisms. Together, these data suggest that loss of CLAMP affects the splicing of multiple components of the sex determination pathway.”

Also, we have highlighted in the discussion that there are multiple potential mechanisms by which the loss of CLAMP could regulate sex-specific splicing in the Discussion section (Lines 499-504) “Our results support a hypothesis that the loss of CLAMP may influence sex-specific splicing through multiple mechanisms: (1) CLAMP directly binds to DNA of a subset of target genes, including the sxl gene itself and other key regulators of alternative splicing; (2) CLAMP influences splicing of other targets indirectly by regulating sxl splicing and Sxl protein levels, and (3) Lastly, we speculate that CLAMP may further influence splicing via additional currently unknown mechanisms such as interactions with RNA and RNA binding proteins involved in splicing with which it associates^14^.”

Another concern was the question of uncoupling the effects of CLAMP on transcription and splicing. The authors provide some comment that 85% of genes affected at the level of splicing are not regulated at the level of transcription, but there is no data shown or any details as to how this comparison was done.

These data were previously shown in a Venn diagram in the supplement, and we have now moved them to main Figure 3.

Overall, the paper has not changed much from the previous version we reviewed. The authors present a very large amount of data. These are not always totally clear and often seem over-interpreted. I still do not think that the strong push for a role of CLAMP as a master regulator of splicing is substantiated.Reviewer #2 (Recommendations for the authors):The authors of this manuscript have added significant additional data to support a role for maternal Clamp in the regulation of sex-specific alternative splicing. These data, while correlative, help to convince me that regulation of alternative splicing is a major function of Clamp. Intriguingly, the new data present yet another mechanism of Clamp-dependent gene regulation through 5'UTR association and FMRP-dependent regulation of translation. This surprising finding helps put to rest a concern from the previous version of the manuscript, where the reduction of Sxl protein levels seemed to poorly correlate with the magnitude of the change in splicing observed. As such, it would seem Clamp's major role in Sxl regulation occurs at the level of translation control, as opposed to splicing or transcription initiation. An additional complication that will warrant future investigation.The manuscript is dense and completely packed full of data and analyses. This strength is also its flaw, as it remains challenging to follow the thread of the story at times as the models and assay systems change. Nevertheless, I feel that it is important that the work should be published without further delay so that others may benefit from the discoveries and data sets described in this work.I have a few suggestions for the authors about ways to help clarify the presentation.1. It would be wonderful if the figure legends would include the identity of the assay used to collect the data. For example, the legend for Figure 4E does a great job of this, but Figure 3, the rest of Figure 4, and Figure 5 would benefit from the same level of detail. This saves the reader from jumping back and forth so much between the text, the figure, and the legend while trying to understand the data.

We thank the reviewer for this critique which will greatly improve the manuscript's clarity. To address reviewer concerns, we have removed Figures 4 and 5; Figure 3 is the new Figure 4. In Figure 4, the number of plots has been reduced to only show average profiles for CLAMP occupancy at CLAMP-dependent sex-specifically spliced genes in females (red line) and in males (blue lines) at 0-2 Hr (pre-MZT) (Figure 4A, C) and 2-4 Hr (post-MZT) (Figure 4B, D) in females (Figure 4A, B) and males (Figure 4C, D). Then, we compared the average CLAMP binding pattern at sex-specifically spliced genes (Figure 4A-D) to the CLAMP binding pattern at genes whose transcription but not splicing is both sex-biased and dependent on CLAMP (Figure 4E-H). Also, we have labeled the number of genes in each group, N, on the figures so that the reader does not have to refer to the supplement.

2. The rationale for using specific statistical tests should be presented in the manuscript and/or legend. Perhaps a section in the methods for statistical analysis? For example, Figure 1D relies upon a chi-square test (why not ANOVA?) while Figure 2A relies upon Fisher's exact test. I think I understand why (sample size), but I'm guessing. Figure 2E the p-value range is not clear for the left and center panels.

Since the Chi-Square test is used when every variable is categorical and the ANOVA when there is at least one categorical variable and one continuous dependent variable, we used Chi-Square Test for 1D because we do not have any one continuous dependent variable.

Methods for statistical analysis are described in each respective methods section under the computational methods section. When a commonly used test was performed, we noted it in each figure legend.

3. Did the authors mean to use "transcriptions" on the left axis of Figure 2A graph or "transcripts"?

Thank you for noting this error which we have corrected as “transcripts.”

4. The 5'UTR in figure 7C should be labeled.

This figure we have removed from the revised version of the manuscript.

Reviewer #3 (Recommendations for the authors):In this manuscript, the authors carry out a detailed examination of sex-specific gene expression and pre-mRNA splicing during early *Drosophila* embryogenesis. They take good advantage of a meiotic drive system that had been previously implemented in the PI's lab to enable the collection of sufficient amounts of properly sexed embryos to perform various genomic assays. The authors carry out transcriptome (RNA-seq) and chromatin (Cut&Run) profiling experiments in the presence or absence of a maternally provided transcription factor, called CLAMP (chromatin-linked adaptor for MSL proteins). They analyze two different developmental time points (0-2 hrs and 2-4hrs after egg laying) corresponding to pre- and post-ZGA (zygotic gene activation) embryos, respectively.This is a huge manuscript (80+ pages) with a ton of supplemental figures and data. There is a lot to like here, but in my view, the manuscript needs further revision. Most of my critiques can be addressed without further experimentation. There is a good story here but I feel that the stronger points get diluted by the weaker arguments.Response to Previous ReviewI did not participate in the previous round of review and so have tried to avoid bringing up new points that were not raised in the first round. Rather than diving into the details straightaway, I would say that the main criticism raised by the referees was one of data over-interpretation. Personally, I am not comfortable making deep mechanistic conclusions (e.g. an association with the catalytic step 2 spliceosome) largely on the basis of genome-level analyses. After reading the revised manuscript and the response to the review I still feel that some of the data are being pushed beyond their limits. The authors' model may well be correct, but the narrative in many parts of the manuscript goes from a given finding being what I would say is "consistent with" a certain interpretation, rather than one that actually "suggests" it works that way.General points1. CLAMP is a general transcription factor, but it has a well-documented role in the histone locus body (HLB), located at the histone gene complex (HisC). Reduced expression of histones can have major effects on gene expression (on both transcription initiation and downstream RNA processing steps). The potential for pleiotropic effects on transcription (e.g. elongation rates are known to affect splicing) due to reduced histone dosage is not really mentioned.

We thank the reviewer for the helpful comments. We agree that reduced histone dosage may cause pleiotropic effects on splicing by altering transcription. Therefore, we have carefully compared and contrasted the effects on splicing and transcription in the absence of CLAMP as follows:

We quantified changes in splicing using PSI values, an exon-centric approach implemented as part of the SUPPA algorithm that measures the percentage of each exon spliced into each transcript and is not isoform-centric. Therefore, we are not simply quantifying changes in the production of each isoform. We compared the CLAMP-dependent sexspecifically spliced genes with CLAMP-dependent differentially expressed genes (Figure 3B) and showed that 84.3% (343/407) of genes that require CLAMP for their sex-specific splicing in early embryos do not require CLAMP for their transcription. In contrast, we found that in third-instar larvae, 60% (151/253) of CLAMP-dependent sex-specifically spliced genes are also regulated at the level of transcription in contrast to 15.7% (64/407) of sex-specifically spliced genes in embryos (Figure 3B, C). Therefore, CLAMP has a dual role in transcription and splicing that differs at different developmental stages (Lines 263273).We have compared and contrasted the binding pattern of CLAMP at genes where the loss of CLAMP alters splicing vs. transcription (Figure 3). We noted a dramatic difference in the binding pattern of CLAMP and its proximity to splice junctions at sex-specifically spliced genes, 84% of which are not regulated at the transcription level.

2. Line 142. Claims of primacy should be removed from the Results section. That sort of thing can be used in an introduction summary or in the discussion of the results. Using it as a conclusion in the Results section ("Therefore, we defined sex-specific splicing events in the early embryo for the first time.") just seems a bit odd.

Thank you for this comment; we deleted this claim in the text.

3. Some aspects of the Results need to be reworked. Probably got mixed up in the revision. As it now stands, the subsection starting on line 150 is redundant and out of sequence. There is a whole list of reasons for doing these experiments in the Intro (lines 78-87), seems like line 150 starts to make the same arguments over again. Furthermore, the information on lines 154-159 really should have been introduced on/near line 128 where the authors first present results of splicing analysis following CLAMP depletion.

Thank you for this critique which we have addressed.

4. Line 188. The way that this sentence is written, the authors have already concluded that CLAMP regulates splicing. At this point in the narrative, loss of maternal CLAMP could affect SSS by any number of indirect means. "Regulation" implies something more active. So I'm not sure you can say start off with: "Furthermore, 85% of genes at which clamp regulates SSS…" because it assumes facts that are not in evidence. I apologize if this comment sounds picayune but this sort of logic matters when you are building an argument.

We have addressed the critique as follows: We have used “CLAMP dependent sex-specifically spliced genes” to define “genes requiring CLAMP for sex-specific splicing” and “CLAMP influences sex-specific splicing.” (Lines 499-504)

5. The Cut&Run data in Figure 6 are curious. I worry that there is some sort of normalization problem with the dataset. In the peaks that were identified in the male control embryos (panel A), roughly two-thirds of the sites on the autosomes and half the sites on the X chromosome are essentially flat. Does that mean the peak that was called by MACS2 is really more than 1.5 kb wide? Maybe that makes some sort of sense on the X. But on the autosomal sites, it looks like noise. The signal in the flanking regions next to the sharper peaks in the second heatmap column (Male control) looks too high. Well above the background binding levels in all of the other columns. This suggests that maybe there are simply more reads in this sample. I cannot tell without seriously digging into it. Is MLE really coating large chunks of chromatin on the autosomes?Why is there such an abrupt transition from the set of narrowly defined peaks to a set of wide, shallow ones? If there were some sort of second criterion one could use for peak calls then you might be able to exclude (or include) certain regions. Right now it just looks like autosomal noise.Again, for the autosomes at least, it might make some sense to try and find a common set of peaks that are found in both the male and female control samples. Then look at the effects of CLAMP depletion on that subset. In addition to a heatmap, one could use DESeq2 to quantify the difference in a metaplot for males vs females (+/- clamp).

We agree with the reviewer that these data need further analysis. Therefore, we have removed these data from the present manuscript and plan to perform additional genomic and microscopy experiments to define CLAMP-MLE interactions in the future.

Specific ideas for revisions:Figure 1. Panels A and D. The multiple uses of various shades of gray in panel D versus similar shades of gray in panel A are confusing. This could be improved with color to make it a bit more readable. In panel A, I suggest that the authors use a unique color for each of the "differential" exons (currently they are all in black) examined in the 7 classes. Then shade the corresponding bars in panel D with that same color (one of which could be black). That way you can continue to use gray in panel A for the exons that are not differentially analyzed.

We have addressed this good suggestion by color-coding panels A and D according to the type of alternative splicing event.

Figure 2E and lines 212-214. Most of the SSS genes in the early female embryo encode transcription and splicing factors? This statement is not at all obvious or even well supported by Figure 2E. There are two dots, one roughly 3 genes and the other 6-7 genes? What is the numerator? What is the denominator? Why should I believe this finding is significant? Why is the adjusted p-value bar all one shade in the two on the left? More stat power in the third GO term panel? I feel like the major points being made in Figure 2A-B get diluted with the additional panels and Venn diagrams. Better to focus the reader on solid conclusions.

Thank you for this suggestion which we have implemented. We replotted Figure 2E using ShinyGO v0.75c: Gene Ontology Enrichment Analysis with an FDR cutoff 0.05. The number of genes in each group (N) is listed at the top of each dot plot. The number of genes within each GO category is noted as the circle size, and GO biological processes are plotted according to the level of fold enrichment along the x-axis.

Lines 239-274. Three full paragraphs of text are assigned to Supplementary figures. Does that not seem excessive? If the RNA-seq data regarding zygotic CLAMP expression are not going to be presented in the main body figures, why so much text? This leads to Figure 3.

We thank the reviewer for this suggestion and have addressed this issue by moving several of the supplementary figures and including one in the main text (Figure 3). This is possible due to the removal of iCLIP and MLE CUT&RUN data.

Figure 3. Important points are being made here, but I fear some of the points are getting lost in the blizzard of metaplots. I don't have any good suggestions for how to streamline but it seems if a few key points could be distilled out of Figures3, and from the supplementary tables cited in the three paragraphs above (lines 239-274), that a single main-body figure with most important points would be impactful.

We address the important reviewer concerns and have streamlined the figures. The old Figure 3. is now Figure 4. The number of plots has been reduced to only show average profiles for CLAMP occupancy at CLAMP-dependent sex-specifically spliced genes in females (red line) and in males (blue lines) at 0-2 Hr (pre-MZT) (Figure 4A, C) and 2-4 Hr (post-MZT) (Figure 4B, D) in females (Figure 4A, B) and males (Figure 4C, D). Then, we compared the average CLAMP binding pattern at sex-specifically spliced genes (Figure 4A-D) to the CLAMP binding pattern at genes whose transcription but not splicing is both sex-biased and dependent on CLAMP (Figure 4E-H).

Figure 4. This whole figure should be reworked and most of it sent to the supplement. Panel E should be deleted. The information content in panels C and D is really low. That leaves A and B. What are the points being made in these panels? I don't think that the authors make much out of the motifs in the text. So the main points.

This figure has been removed from the revised manuscript.

Figure 5 (see general points above). Reanalysis seems to be in order.

This figure has been removed from the revised manuscript.

Figure 6. This figure may need some reshuffling or even split in two. I think that most of the readers of the paper will be confused by the fact that the males do not show any male-specific splicing in panel B. After reading the nuanced text (lines 519-520) a couple of times, where the authors mention that the embryos have not yet become "fully specified," I realized that this is actually the expected result. Maybe the authors should lead with that. Or be more explicit. In fact, I'm not even sure the pre-ZGA transcripts of Sxl are even translated in the early embryo. But that's a story for another time.In this part of the manuscript, I don't think the reader is quite ready for panel A, which could be combined with panels D and E to make a new figure. Meanwhile, it might be helpful to bring back two of the panels from the older version of this figure (currently in Figure S11). The gene model in the current panel C (X-axis) is poorly annotated and the Y-axis is unlabelled. I found the panel particularly unhelpful and had to look at Figure S11 to figure out what was going on. I suggest bringing back panels S11A and S11F into the main body somehow. This would show the casual reader that later on in development splicing works the way it is depicted in all the textbooks. Then explain to the reader that early embryos have, by definition, maternally spliced Sxl transcripts. The authors have RNA-seq data that for these time points, why not use them? Analysis of splice-junction reads in the RNA-seq could be added to flesh out an entire figure about Sxl splicing.A new figure could be used to show the pathway and the splicing of Tra, Dsx, Msl2, etc.

This is a great suggestion, and we have now reorganized the figures. Figure 6 and Figure S11 are now (1) Figure 5: Alternative splicing of *sxl* transcript and Sxl protein levels is modulated by CLAMP in females, (2) Figure 6: CLAMP-dependent alternative splicing of components of the sex determination pathway

Figure 7. The overall model. I don't know what to say about the current Tfigure other than it's pretty complicated. The biology is complex, so I get it. But am worried that the authors' main points are going to be lost on a readership that will not appreciate all the nuances.

This Figure has been removed from the revised present version of the manuscript.